# Mutation of *CMTR2* in Lung Adenocarcinoma Alters RNA Alternative Splicing and Reveals Therapeutic Vulnerabilities

RNA splicing dysregulation has emerged as a hallmark of cancer and a promising therapeutic target; however, its full landscape in human solid cancer remains poorly characterized. To address this, we perform alternative splicing analyses using RNA-sequencing data from 751 lung adenocarcinoma samples from our cohort integrated with 519 samples from The Cancer Genome Atlas. Visualization of splicing patterns using t-distributed stochastic neighbor embedding reveals substantial inter-tumor heterogeneity driven by distinct molecular subtypes and histological differentiation. We identify a unique molecular subtype associated with inactivating mutations in *CMTR2*, which encodes Cap-specific mRNA (nucleoside-2'-O-)-methyltransferase 2. *CMTR2* mutations are observed in 3.8% of cases and are predominantly truncating mutations, which form an isolated cluster within the splicing landscape. Intrinsic and CRISPR-Cas9-engineered *CMTR2* mutations disrupt alternative splicing and sensitize cancer cells to sulfonamide-based RNA splicing modulators and immune checkpoint blockade therapy. Retrospective patient data confirm the increased sensitivity of *CMTR2*-deficient tumors to immune checkpoint blockade therapy. These findings uncover a previously unrecognized RNA splicing deficiency in human cancers and define a molecular subtype of lung adenocarcinoma driven by RNA splicing dysregulation, suggesting targets for therapeutic intervention in lung cancer.

Dysregulation of RNA splicing recently emerged as a hallmark of cancer[1]. Alternative splicing (AS), a critical process in gene expression, enables the generation of multiple mRNA and protein isoforms from a single gene, thereby supporting diverse gene functions and regulating key cellular processes. This intricate regulatory mechanism governs numerous biological functions, including tissue-specific differentiation, metabolism, neuronal development, and the pathogenesis of various diseases[2]. In malignancies, dysregulated RNA splicing drives tumorigenesis by increasing cell proliferation, inhibiting apoptosis, increasing migratory capacity, and promoting metastasis[1].

The RNA splicing process is orchestrated by the spliceosome, a large RNA-protein complex composed of small nuclear ribonucleoproteins (snRNPs) and various splicing factors. In cancer, splicing dysregulation frequently arises due to mutations in splicing factor genes[3,4]. In hematological malignancies, somatic mutations in genes such as *SF3B1*, *U2AF1*, and *SRSF2*, which are collectively referred to as "spliceosomal mutations", are major drivers of global RNA splicing alterations[1,3,4]. Genome-wide studies have also identified spliceosomal mutations in non-hematological cancers. For example, we and others reported somatic mutations in the splicing factor genes *RBM10* and *U2AF1* in lung adenocarcinoma (LADC), the most prevalent histological subtype of non-small cell lung cancer (NSCLC), with mutation frequencies of approximately 7–8% and 3%, respectively[5–7]. Despite these discoveries, the full landscape of AS in lung cancer and the specific contributions of spliceosomal mutations remain poorly understood.

✉ e-mail: tkkohno@ncc.go.jp; tnakaoku@ncc.go.jp

In this study, we mapped the comprehensive AS landscape in LADC and explored its therapeutic implications. We analyzed RNA-sequencing (RNA-seq) datasets from 751 LADC cases from the National Cancer Center (NCC) Hospital, Japan and 519 cases from The Cancer Genome Atlas Program (TCGA, https://www.cancer.gov/ccg/research/genome-sequencing/tcga) and visualized splicing heterogeneity across the cohorts. We performed t-distributed stochastic neighbor embedding (t-SNE) analysis of a high-dimensional percent spliced-in (PSI) value matrix for skipped exon (SE) events[8] and found distinct LADC subtypes, defined by global RNA splicing patterns. These subtypes were associated with spliceosomal mutations in *RBM10* and *U2AF1* and correlated with histological differentiation status. We identified a LADC subtype characterized by loss-of-function mutations in the Cap Methyltransferase 2 (*CMTR2*) gene. *CMTR2* encodes Cap-specific mRNA (nucleoside-2′-O-)-methyltransferase 2, a key enzyme involved in methylation of mRNA and small nuclear RNA (snRNA)[9,10]. A previous report demonstrated that *CMTR2* is significantly mutated in LADC and is enriched with loss-of-function mutations[11]. Functional analyses using both tumor-intrinsic and CRISPR-Cas9-engineered *CMTR2*-mutant cells showed that *CMTR2* deficiency induces global alterations in RNA splicing and increases vulnerability of the splicing machinery. This heightened sensitivity renders *CMTR2*-deficient cells particularly susceptible to sulfonamides, which are compounds that degrade the splicing factor RNA-binding protein 39 (RBM39)[12,13]. We demonstrated that *CMTR2*-deficient LADCs exhibit increased sensitivity to immune checkpoint blockade (ICB) therapy, both in a syngeneic mouse model and in LADC patients.

## Results

### Comprehensive profiling of alternative RNA splicing in lung cancer

To characterize the landscape of alternative RNA splicing events in lung cancer, we analyzed RNA-seq data from 1017 lung cancer cases in the NCC cohort using the robust rMATS computational framework (Fig. 1a). This tool enables comprehensive identification of AS events across multiple gene sets, including SEs, alternative 5′ or 3′ splice sites, mutually exclusive exons, and retained introns (Fig. 1b)[14,15]. rMATS quantifies the relative inclusion of alternative exons, splice sites, or introns and reports these as PSI values[8,14].

Previous comparative analyses of tumor and normal tissue samples in various cancer types highlighted an increased prevalence of SE events in tumor samples, and this difference is particularly pronounced in lung cancer[16]. To visualize splicing heterogeneity in lung cancer, we applied t-SNE dimensionality reduction to the high-dimensional PSI value matrix of SE events. Histological subtypes with neuroendocrine differentiation, such as small cell lung cancer (SCLC) and large cell neuroendocrine carcinoma (LCNEC), formed closely clustered groups. Adenocarcinoma (ADC, also referred to as LADC) and squamous cell carcinoma (SCC) tended to show different distributions. Some cases showed overlap between these two histological types, possibly representing combined-type samples with each differentiation component. Additionally, pleomorphic carcinoma was scattered throughout the landscape, reflecting its nature as a histological type that contains sarcomatoid and various histological subtype components (Supplementary Fig. 1a). These observations were corroborated by unsupervised hierarchical clustering, which revealed unique splicing patterns across different histological subtypes (Supplementary Fig. 1b).

We next focused on LADC (*n* = 751), the most prevalent histological subtype of lung cancer. Initial analysis of whole-exome sequencing data revealed no clear clustering of cases based on driver oncogene mutations (Fig. 1c), suggesting that factors beyond driver oncogenes primarily contribute to the observed inter-tumor heterogeneity in AS. Somatic mutations in splicing factor genes such as *RBM10* and *U2AF1* have been reported in LADC[5,6]. In this cohort, *RBM10*

and *U2AF1* mutations were detected in 78 (10.4%) and 8 (1.1%) cases, respectively. Cases harboring these mutations formed distinct t-SNE clusters (Fig. 1d, e). Specifically, the *U2AF1* cluster exclusively comprised cases with the S34F mutation, a hotspot mutation in the first zinc finger domain of *U2AF1*[5,6] (Fig. 1d and Supplementary Fig. 1c). The *RBM10* cluster predominantly included cases with truncating mutations, underscoring the role of *RBM10* as a tumor suppressor[17] (Fig. 1e and Supplementary Fig. 1d).

The AS landscape supported the histopathological heterogeneity of LADC. For instance, a t-SNE cluster enriched for *KRAS*-mutant cases exhibited high expression of both *HNF4A* and *MUC5AC*, which are expressed at high levels in invasive mucinous adenocarcinoma, a distinct histological subtype of LADC[18] (Fig. 1c and Supplementary Fig. 1e, f). This cluster frequently displayed loss-of-function mutations in *NKX2-1* (Supplementary Fig. 1g), which lead to loss of pulmonary identity and conversion to a gastrointestinal lineage[19] (Supplementary Fig. 1h). We also examined the epithelial-mesenchymal transition (EMT), a critical histological transformation process involved in cancer progression[20]. Using an established scoring method based on the EMT gene signature[15,20], we assessed EMT status across the t-SNE landscape and found that samples with mesenchymal features predominantly clustered in specific regions and had a different distribution compared with samples with epithelial features (Supplementary Fig. 2a–c). This finding aligns with the known involvement of AS in EMT regulation[21]. A small cluster exhibited characteristics of neuroendocrine differentiation, as indicated by a high neuroendocrine score based on the neuroendocrine gene signature[22,23] (Supplementary Fig. 2d). This differentiation was further characterized by increased expression of *ASCL1* and concurrent upregulation of *SRRM4*, which encodes a splicing factor that drives neuroendocrine differentiation in prostate cancer[24] (Supplementary Fig. 2e, f). Characterization of the alternative RNA splicing landscape revealed distinct cancer-related clusters shaped by splicing heterogeneity and associated with both spliceosomal mutations and histopathological features.

### Identification of *CMTR2* as a putative tumor suppressor gene with smoking-associated mutations in LADC

Examination of the t-SNE landscape alongside clinical data identified a distinct cluster of 15 cases in the NCC LADC cohort that was enriched with smokers (Fig. 1f). Thirteen of these cases (86.7%) harbored mutations in the *CMTR2* gene (Supplementary Fig. 3a and Supplementary Table 1). These mutations, particularly truncating mutations, were significantly overrepresented in this cluster (Fig. 1g). A comparable cluster was identified in the TCGA LADC dataset (*n* = 519)[25], in which 12 of 17 cases with truncating or splice-site *CMTR2* mutations formed a distinct group (Supplementary Fig. 3b and Supplementary Table 2).

*CMTR2* encodes Cap Methyltransferase 2, an S-adenosyl-L-methionine (SAM)-dependent methyltransferase involved in the 5′ capping of RNA. In mRNA, the 5′ cap structure consists of N7-methylguanosine (m7G), whereas in snRNA, it consists of N2,2,7-trimethylguanosine (m2,2,7 G); both are linked to the first transcribed nucleotide via a unique 5′−5′ triphosphate bond. In vertebrates, CMTR1 and CMTR2 proteins methylate the first and second nucleotides at the 2′-O-ribose position, forming Cap1 and Cap2 RNA modifications, respectively[9,10,26] (Fig. 1h). The Cap2 modification, which is present in approximately 50% of polyadenylated RNA in human cells, plays a crucial role in RNA stability, RNA processing, and immune evasion[27]. *CMTR2* is essential for mammalian embryonic development, as demonstrated by the embryonic lethality observed in *Cmtr2*-knockout mice[28,29]. Despite its essential role in RNA metabolism, the contribution of CMTR2 to RNA splicing and tumorigenesis remains largely unexplored.

To expand upon these findings, we examined TCGA PanCancer Atlas datasets[25] and identified *CMTR2* mutations at notable frequencies

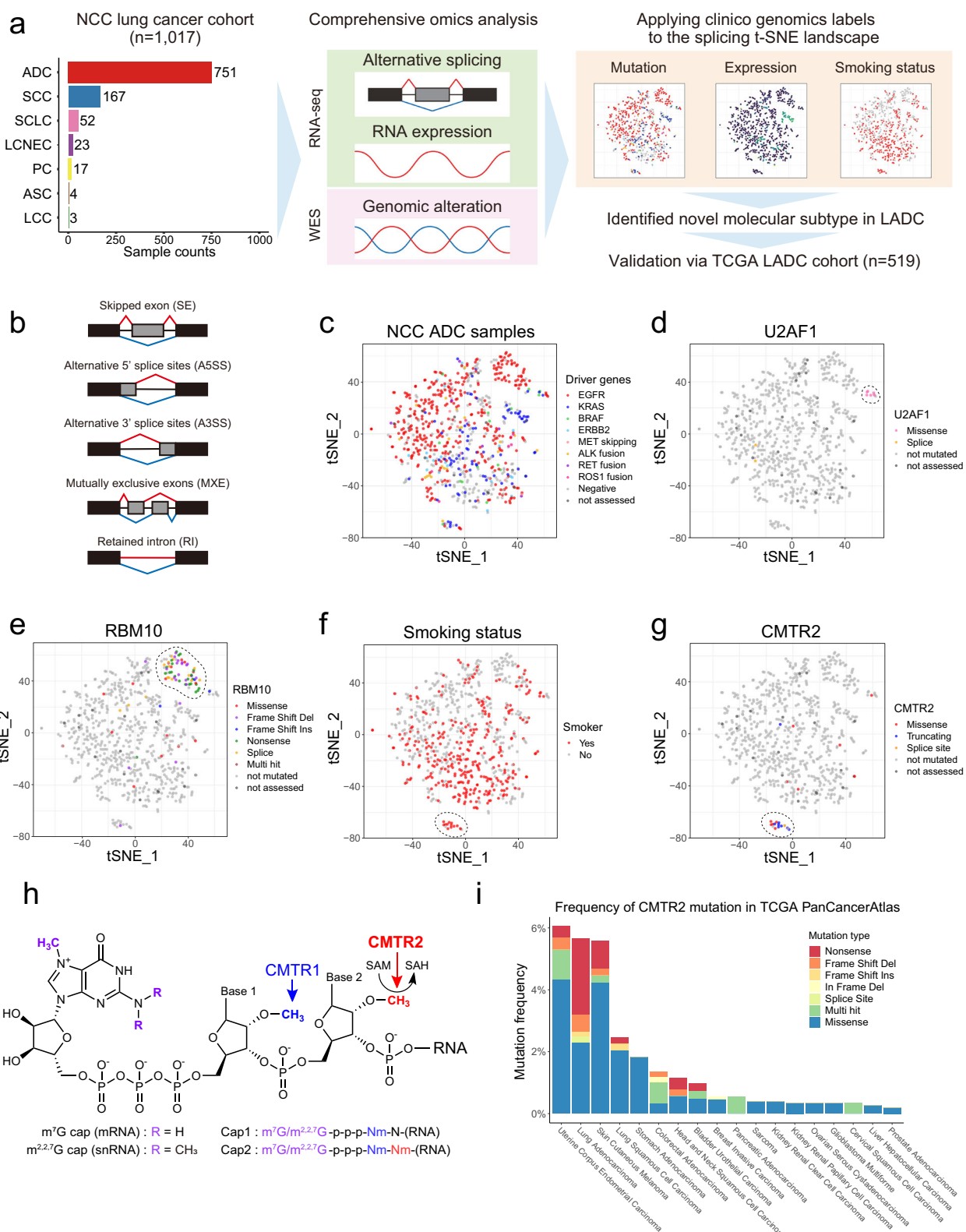

**Fig. 1 | CMTR2 cluster in the AS landscape of LADC. a** Schematic showing the study workflow. The bar graph on the left shows the number of tumor samples of each histological type in the NCC lung cancer cohort. Abbreviations: ADC, adenocarcinoma; ASC, adenosquamous carcinoma; LCC, large cell carcinoma; LCNEC, large cell neuroendocrine carcinoma; PC, pleomorphic carcinoma; SCC, squamous cell carcinoma; SCLC, small cell lung carcinoma. **b** Types of AS events detected by rMATS. **c**–**g** AS landscape of LADC samples from the NCC cohort, analyzed based on SE PSI values. Each t-SNE plot reflects the reduction in the higher-dimensional splice event PSI matrix for each sample into two dimensions. t-SNE plots are color-coded according to driver genes (**c**), type of *U2AF1* mutation (**d**), type of *RBM10* mutation (**e**), smoking status (**f**), and type of *CMTR2* mutation (**g**). **h** RNA cap structure. Methylation sites and the corresponding methyltransferases are indicated. The chemical structure was modified from CHEBI:167614 obtained from the ChEBI database (https://www.ebi.ac.uk/chebi/). **i** Frequency of each *CMTR2* mutation type in the TCGA PanCancer Atlas.

in uterine corpus endometrial carcinoma (32/529 cases, 6.0%), LADC (32/566 cases, 5.7%), and cutaneous melanoma (25/448 cases, 5.6%) (Fig. 1i). In cancers with relatively high *CMTR2* mutation frequencies, the mutations were often associated with hypermutated subtypes characterized by *POLE* mutations or microsatellite instability-high (MSI-H). However, this association was not observed in lung cancer (Supplementary Fig. 3c). To further investigate the role of *CMTR2* mutations in cancer, we used the IntOGen web platform[30], which integrates data from diverse cancer sequencing projects and employs multiple in silico methods to identify potential driver mutations. *CMTR2* was identified as a putative driver gene in two independent lung cancer cohorts (TCGA LADC and the Hartwig Medical Foundation NSCLC cohort[31]) but not in other cancer types (Supplementary Table 3). These results suggest that while *CMTR2* mutations may be passenger mutations in hypermutated subtype cancers, they appear to have functional significance in lung cancer.

Based on these findings, we reanalyzed TCGA LADC data using programs included in the IntOGen workflow. Both the MutPanning[32] and OncodriveFML[33] programs identified *CMTR2* mutations as driver mutations alongside other well-established genes implicated in LADC (Supplementary Fig. 4a–c). Clonal population analysis using PyClone-VI[34] demonstrated that *CMTR2* mutations were truncal and co-occurred with key driver mutations such as those in *KRAS* and *TP53* in the NCC LADC cohort (Supplementary Fig. 4d, e). The proportion of truncating *CMTR2* mutations, especially nonsense mutations, was significantly higher in LADC (2.5%) than in other cancer types (Fig. 1i). These truncating mutations were frequently associated with reduced copy numbers at the *CMTR2* locus, indicating loss of the wild-type (WT) allele (Supplementary Fig. 4f). In lung cancer, stop-gain (nonsense) mutations are commonly observed in key tumor suppressor genes such as *TP53* and are strongly enriched in mutational signatures linked to tobacco smoking[35]. Combined analysis of the NCC and TCGA cohorts revealed that *CMTR2* mutations were significantly more frequent in smokers than in non-smokers (5.5% *vs.* 0.8%, respectively; Supplementary Table 4). Furthermore, most *CMTR2* mutations were C > A transversions, consistent with a mutational signature (SBS4) strongly associated with smoking[36] (Supplementary Fig. 4g). In summary, these findings provide insight into the role of *CMTR2* mutations in LADC and confirm previous reports of *CMTR2* as a putative tumor suppressor gene, adding the observation that *CMTR2* mutations are mostly smoking-associated.

### Deleterious effects of missense mutations in *CMTR2*

In the NCC cohort, the t-SNE cluster harboring *CMTR2* mutations included cases with missense mutations (Fig. 1g). The CMTR2 protein has two major domains: the N-terminal catalytic Rossmann-fold methyltransferase (RFM) domain and the C-terminal non-catalytic RFM domain[37]. In the catalytic domain, the K-D-K triad motif, comprising residues K117, D235, and K275, is critical for Cap2 methylation activity[9]. Among the observed missense mutations, two variants, K117N and K275N, were located in this motif (Fig. 2a, b). The critical role of the K-D-K triad in methylation[9] strongly suggests that these mutations disrupt the enzymatic activity of the CMTR2 protein. The L291R mutation, which was recurrent in cases within the *CMTR2* t-SNE cluster, was identified as a truncal mutation (Fig. 2a and Supplementary Fig. 4e). This finding indicates that missense mutations outside the K-D-K motif may also impair CMTR2 function.

To investigate the structural impact of these mutations, we performed molecular dynamics (MD) simulations using the supercomputer Fugaku[38] and analyzed a conformational difference between the WT CMTR2 protein and the K117N or K275N mutant, focusing on a region consisting of the K-D-K triad, mRNA/snRNA 5' cap structures, and SAM. Our simulations revealed that the geometry of an interaction network between the catalytic motif and substrate components was

altered in both mutants. The mutants exhibited increased distances between the catalytic residues and the mRNA/snRNA second transcribed nucleotide, providing insight into the molecular mechanism underlying the disrupted enzymatic activity (Fig. 2c and Supplementary Fig. 5a–c).

Furthermore, to systematically evaluate the pathogenicity of all identified *CMTR2* missense mutations, we used AlphaMissense (AM) to predict their functional impact[39]. Missense variants found in cases that formed t-SNE clusters in the NCC or TCGA cohort consistently exhibited high pathogenicity scores, whereas variants outside these clusters tended to exhibit lower scores (Fig. 2d, e and Supplementary Tables 1, 2). High-scoring missense mutations were predominantly located in the catalytic RFM domain, although some, such as the S508P mutation in the NCC cohort, were located in the non-catalytic domain (Fig. 2a, b, d). Truncated proteins lacking the non-catalytic domain region lack methylation function[9]. These findings suggest that the non-catalytic domains play an auxiliary but essential role in supporting the methylation activity of CMTR2. Thus, it is plausible that missense mutations in the non-catalytic domain also impair the enzymatic function of CMTR2. Taken together, these results demonstrate that missense mutations in *CMTR2* have deleterious effects on the function of this enzyme, further supporting the hypothesis that *CMTR2* acts as a tumor suppressor gene.

### RNA splicing alterations caused by CMTR2 deficiency

The role of *CMTR2* mutations in RNA splicing in cancer has not been investigated to date. We evaluated the impact of these mutations on AS. In both the NCC and TCGA cohorts, when normal tissue splicing data were included in the t-SNE analysis, normal and tumor tissues were clearly segregated (Fig. 3a and Supplementary Fig. 6a). Quantification of differential splicing events between tumor clusters and normal tissues showed that clusters harboring mutations in *U2AF1*, *RBM10*, and *CMTR2* had significantly more splicing alterations than WT tumors (Fig. 3b and Supplementary Fig. 6b). Additionally, similar to the mutual exclusivity of major splicing factor mutations in solid and hematological malignancies[40], *CMTR2* mutations rarely co-occurred with *RBM10* and *U2AF1* mutations in both cohorts (Supplementary Fig. 7a, b and Supplementary Tables 1, 2). These findings suggest that *CMTR2* mutations induce splicing changes similar to those caused by spliceosomal mutations[4].

In myeloid malignancies and lung cancer, spliceosomal mutations often lead to specific aberrant splicing patterns. For example, *SF3B1* mutations primarily result in reduced intron retention and differential 3' splice site usage, whereas *SRSF2* and *U2AF1* mutations mainly induce alternative SE events[41–43]. Consistent with previous reports, splicing events that were markedly altered in *U2AF1*-mutated cases in the NCC and TCGA cohorts were enriched for SE events (Fig. 3c and Supplementary Fig. 6c). Similarly, SE events dominated the splicing alterations observed in *CMTR2*-mutated clusters in both cohorts (Fig. 3d and Supplementary Fig. 6d).

To directly test whether *CMTR2* deficiency drives these splicing alterations, we generated biallelic *CMTR2*-knockout cell lines from two human LADC cell lines, A549 and NCI-H3122, using CRISPR-Cas9. Three independent A549 and two independent NCI-H3122 *CMTR2*-knockout clones were established (Supplementary Fig. 8a–d). Additionally, *CMTR2*-knockout cells were generated from the mouse Lewis lung carcinoma (LLC) cell line (Supplementary Fig. 8e). RNA-seq analysis of these clones revealed widespread splicing alterations predominantly involving SE events, mirroring the splicing changes observed in *CMTR2*-mutated LADC cases from the NCC and TCGA cohorts. Thousands of significantly altered SE events were observed in *CMTR2*-knockout cells compared with control cells (Fig. 3e, f and Supplementary Fig. 8f, g). Principal component analysis (PCA) of SE events according to PSI values robustly separated *CMTR2*-knockout cells from control cells (Fig. 3g and Supplementary Fig. 8h). These

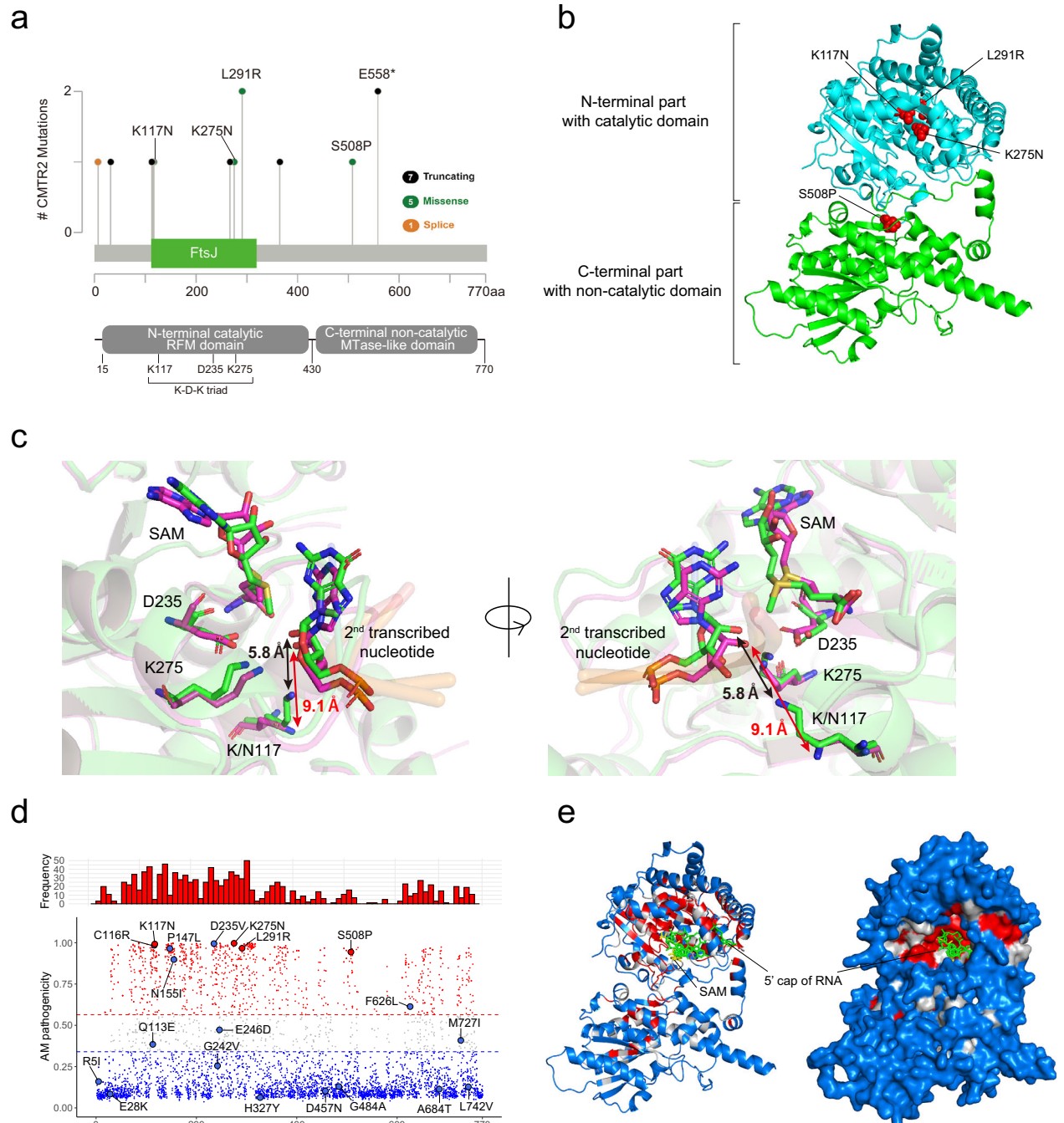

**Fig. 2 | Distribution of *CMTR2* missense mutations within the NCC cohort.**
**a** Top: Lollipop plot showing *CMTR2* mutations identified in the CMTR2 cluster in the NCC cohort. Bottom: Domain structure of the CMTR2 protein aligned with the amino acid positions shown in the lollipop plot above. **b** Predicted CMTR2 protein structure. The N-terminal region harboring the catalytic domain is shown in cyan and the C-terminal region harboring the non-catalytic domain is shown in green. The *CMTR2* missense mutations in the CMTR2 cluster of the NCC cohort are shown in red. **c** Superimposition of the mean structures obtained from five 1-μs MD simulations of WT CMTR2 (green) and the K117N mutant (magenta) complexed with an mRNA substrate and SAM. The main chains of the protein and mRNA are shown as a transparent ribbon model, while the catalytic K-D-K motif residues (K/ N117, D235, and K275), SAM, and the second transcribed nucleotide of the mRNA

are highlighted with stick models. Black and red arrows indicate the distances between the amino nitrogen of K/N117 and the 2'-O position of the second transcribed nucleotide, respectively. **d** All possible missense mutations in *CMTR2*, with AM pathogenicity scores (y-axis) plotted against amino acid positions (x-axis). Large dots represent *CMTR2* mutations in the TCGA or NCC LADC cohort; mutations in cases within and outside the CMTR2 cluster in the t-SNE plot are shown in red and blue, respectively. The bar chart above the scatter plot shows the frequency of likely pathogenic mutations at each amino acid position. **e** Predicted protein structure of CMTR2 with the 5′ RNA cap (green) and SAM (yellow). Left: cartoon; right: surface representation. The color of each residue represents the average AM pathogenicity score. Residues are shown in red (likely pathogenic), blue (likely benign), or gray (ambiguous).

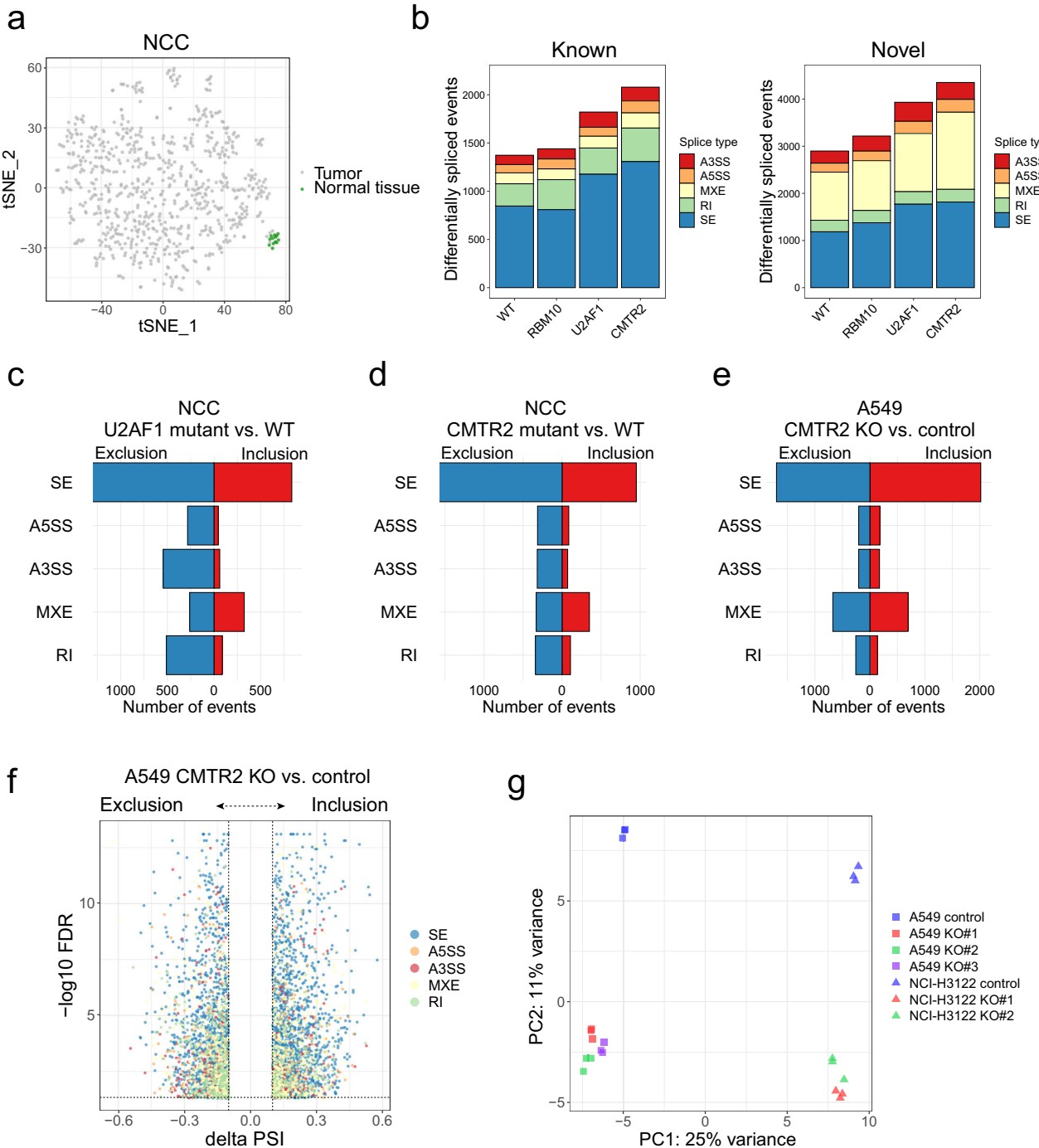

**Fig. 3 | RNA splicing alteration caused by *CMTR2* deficiency. a** t-SNE plots for LADC (tumor) samples and normal tissues from the NCC cohort analyzed according to SE PSI values. **b** Stacked bar charts indicating the count and type of differentially spliced events identified by comparing normal tissues with WT tumors and *RBM10*-, *U2AF1*-, and *CMTR2*-mutated samples (with only truncating mutations in *CMTR2*); *n* = 5 each. Data were randomly sampled from each cluster within the NCC cohort. Spliced events were defined as those with |ΔPSI| ≥ 0.1 and an FDR < 0.05. Left: known (annotated) spliced events. Right: novel (unannotated) spliced events based on known or novel splice sites. **c**–**e** Bilateral bar charts depicting the number of significant differentially spliced events between *U2AF1* (c) or *CMTR2* (d) mutant and WT tumors in the NCC cohort (*n* = 11 per group; *CMTR2* mutant samples were selected from those within the CMTR2 cluster that did not harbor a combination of other splicing factor mutations) and between *CMTR2*-knockout and control A549 cells (e). FDR < 0.05; red: inclusion events (ΔPSI ≥ 0.1); blue: exclusion events (ΔPSI ≤ −0.1). **f** Volcano plot showing differentially spliced events induced by *CMTR2* knockout in A549 cells. Cutoff: |ΔPSI| ≥ 0.1; FDR < 0.05. **g** PCA plot showing differences in AS between *CMTR2*-knockout clones and control cells. Each dot represents one of three replicates for each clone or control cells. PCA analysis was performed using the PSI value of each cell.

results demonstrate that *CMTR2* loss-of-function mutations induce significant splicing alterations, among which SE events are the most prominent.

## Splicing events affected by CMTR2 deficiency
Next, we examined the overlap of differentially spliced genes between *CMTR2*-mutated cases from the NCC and TCGA cohorts and *CMTR2*-

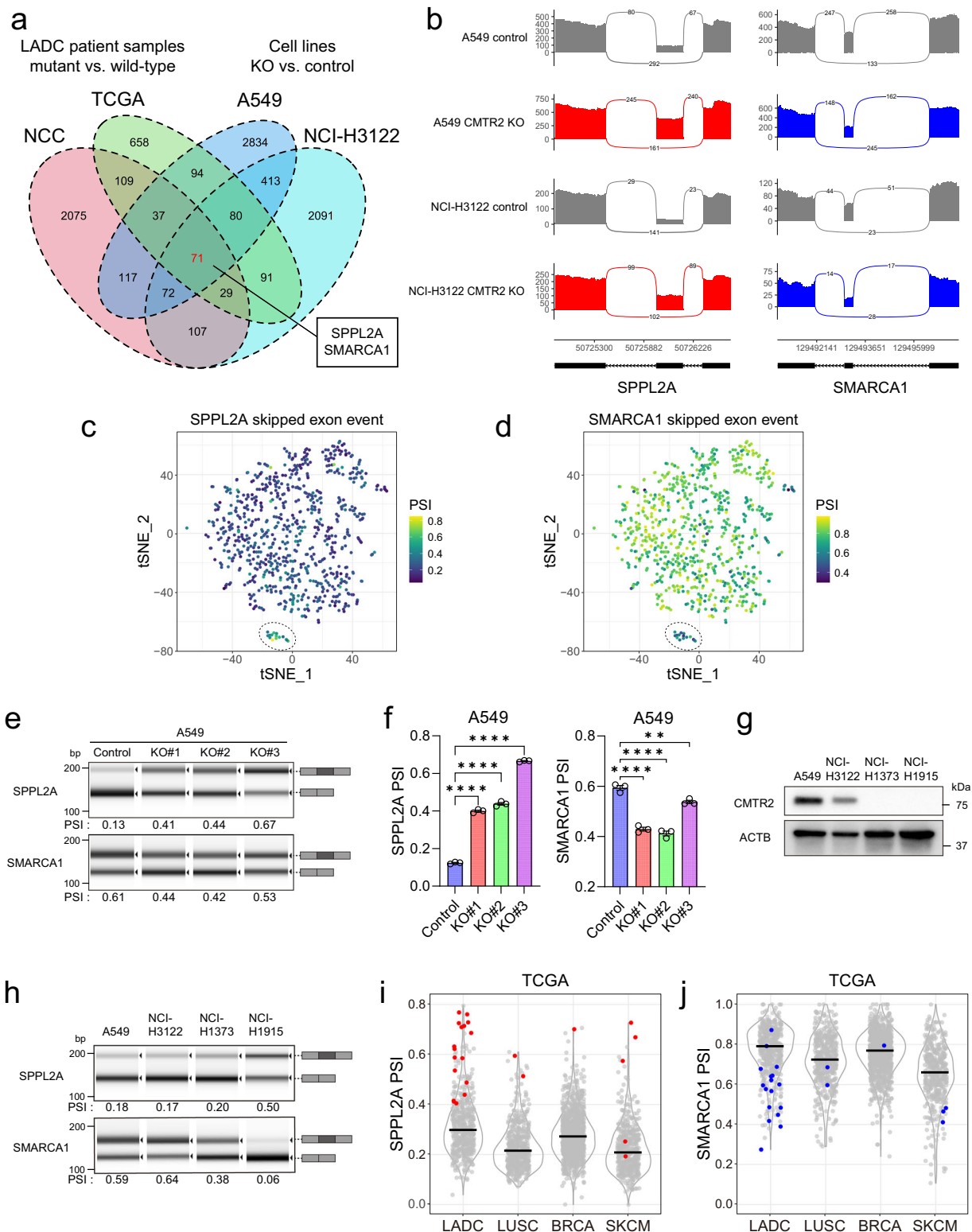

knockout cell lines. Seventy-one SE events were common to both human tumor samples and knockout cell lines (Fig. 4a, Supplementary Fig. 9a, and Supplementary Data 1). Among these shared splicing events, *SPPL2A* (Signal Peptide Peptidase-Like 2 A) and *SMARCA1* (SWI/SNF-Related, Matrix-Associated, Actin-Dependent Regulator of Chromatin, Subfamily A, Member 1) were selected for further validation due to their robust PCR-detectable exon length; *SPPL2A* showed increased exon inclusion and *SMARCA1* showed increased exon exclusion (Fig. 4b and Supplementary Fig. 9b–e). In the t-SNE AS landscape of the NCC

cohort, these splicing alterations were observed specifically in *CMTR2*-mutated cases (Fig. 4c, d). Furthermore, we performed long-read direct RNA-seq of A549 *CMTR2*-knockout cells, which demonstrated changes in read coverage consistent with short-read sequencing (Supplementary Fig. 10a–d). Isoform quantification analysis also revealed altered ratios between exon-including and exon-skipping isoforms of these genes (Supplementary Fig. 10e–g). These splicing changes were validated by performing RT-PCR analysis of both *CMTR2*-knockout cell lines and the NCI-H1373 and NCI-H1915 lung cancer cell

**Fig. 4 | Reproducible SE events induced by *CMTR2* deficiency. a** Venn diagram showing the overlap of significantly differential SE events in *CMTR2* mutant *versus* WT samples in the TCGA and NCC cohorts and in *CMTR2*-knockout *versus* control A549 and NCI-H3122 cell lines. **b** RNA-seq read coverage plots showing differential SE events in *SPPL2A* (left) and *SMARCA1* (right) in A549 and NCI-H3122 cells upon *CMTR2* knockout. **c, d** t-SNE plots of LADC samples from the NCC cohort were analyzed based on SE PSI values and color-coded according to the SE PSI of *SPPL2A* (c) and *SMARCA1* (d). **e** Agilent TapeStation gel-like images of RT-PCR products validating differentially spliced events in *SPPL2A* and *SMARCA1* upon *CMTR2* knockout in A549 cells. **f** Quantification of PSI values for *SPPL2A* (left) and *SMARCA1* (right). *n* = 3 independent experiments per condition. Error bars, ± SEM. Statistical significance was determined by a one-way ANOVA followed by Dunnett's multiple comparison test. *SPPL2A*: ****$p < 0.0001$; *SMARCA1*: **$p = 0.0043$, ****$p < 0.0001$. **g** Western blot analysis of CMTR2 expression in NSCLC cell lines (representative of three independent experiments). **h** Comparison of *SPPL2A* and *SMARCA1* splicing patterns in CMTR2-expressing A549 and NCI-H3122 cells and CMTR2-deficient NCI-H1373 and NCI-H1915 cells (representative of three independent experiments). **i, j** Violin plots depicting PSI values for *SPPL2A* (i) and *SMARCA1* (j) SE events across different tumor types from the TCGA cohort: LADC, lung adenocarcinoma (*n* = 518); LUSC, lung squamous cell carcinoma (*n* = 499); BRCA, breast invasive carcinoma (*n* = 1044); and SKCM, skin cutaneous melanoma (*n* = 462). Red and blue plots indicate samples harboring *CMTR2* truncating mutations. Black lines indicate median values.

lines, which harbor truncating *CMTR2* mutations (Fig. 4e–h, Supplementary Fig. 11a–c, and Supplementary Table 5). Overexpression of WT *CMTR2* in the NCI-H1373 cell line decreased the altered SE events (Supplementary Fig. 12a-c).

Exon inclusion in *SPPL2A* and exon exclusion in *SMARCA1* were observed not only in LADC, but also in MDA-MB-231 cells, a breast cancer cell line with a truncating *CMTR2* mutation (Supplementary Fig. 13a, b and Supplementary Table 5). Increased exon inclusion in *SPPL2A* and increased exon exclusion in *SMARCA1* were observed in multiple cancer types in the TCGA dataset (Fig. 4i, j). These findings suggest that *CMTR2* deficiency alters the splicing of diverse genes and that certain genes are commonly affected in different cellular contexts.

## CMTR2 interactions with splicing machinery

We next investigated the mechanistic link between CMTR2 and splicing dysregulation. Analysis of protein–protein interaction databases[44,45] indicated there are physical interactions between CMTR2 and spliceosomal complex components, particularly snRNPs (Supplementary Tables 6, 7). To validate this, we generated Flp-in T-REx 293 cells expressing FLAG-tagged WT CMTR2 or the K117N mutant, a loss-of-function variant identified in our NCC cohort, under doxycycline control (Fig. 5a, b). FLAG immunoprecipitation was performed to isolate CMTR2 and its interacting proteins. SDS-PAGE separation of immunoprecipitates revealed protein bands specific to WT CMTR2 samples (Fig. 5c). Mass spectrometry analysis of these bands identified enriched U1 snRNP proteins in WT CMTR2 complexes, confirming that the association of U1 snRNP with the K117N mutant was impaired (Fig. 5d and Supplementary Data 2). Western blotting of immunoprecipitated complexes confirmed that the U1 snRNP component SNRNP70 did not bind to the K117N mutant (Fig. 5e).

These results suggest that *CMTR2* deficiency potentially affects splicing through mechanisms related to physical interactions between CMTR2 and snRNPs (Fig. 5f). Consistently, the 71 shared SE events exhibited GT-AG splice sites characteristic of major introns targeted by U1 snRNP[1] (Supplementary Fig. 14).

To further explore the impact of *CMTR2* deficiency on the splicing machinery, we performed differential gene expression analysis followed by Gene Set Enrichment Analysis (GSEA) of *CMTR2*-knockout and control A549 cells. This analysis revealed that core genes involved in spliceosomal complexes, such as those encoding snRNP subunits, SR proteins, hnRNPs, the PRP19 complex, and LSm proteins, were significantly upregulated in *CMTR2*-knockout cells (Fig. 5g and Supplementary Fig. 15a, b). Spliceosome-related gene enrichment was also observed in murine LLC cells, with substantial overlap of core enriched genes between species (Supplementary Fig. 15b, c).

These findings demonstrate that CMTR2 interacts with components of the splicing machinery, indicating that *CMTR2* deficiency affects broad splicing networks.

## CMTR2 deficiency and sensitization to splicing inhibition

Next, we investigated whether *CMTR2* deficiency is associated with potential therapeutic vulnerabilities in lung cancer. We analyzed drug sensitivity data from the DepMap portal[46], which includes information on a wide range of human cancer cell lines, including two NSCLC lines (NCI-H1373 and NCI-H1915) with truncating mutations in *CMTR2*. A search for drugs with low area under the curve (AUC) values in these cell lines identified indisulam, which displayed a notably low AUC value among 57 NSCLC cell lines (Fig. 6a and Supplementary Fig. 16a). Indisulam is a sulfonamide-based anticancer agent that targets and degrades the splicing factor RBM39[12]. RBM39 degradation by indisulam induces splicing abnormalities by disrupting the RNA-binding protein interactome, which includes several proteins involved in mRNA splicing, and causes synthetic lethality in tumors with spliceosomal mutations[47]. Thus, our findings suggest that *CMTR2* deficiency is a potential therapeutic target for RBM39 degraders. Indeed, indisulam induced apoptosis and growth suppression specifically in *CMTR2*-deficient cells, although it degraded RBM39 protein independently of *CMTR2* status (Supplementary Fig. 16b, c). Indisulam also increased exon skipping and intron retention in these two cell lines, which is consistent with previous reports[12] (Supplementary Fig. 16d–g).

To validate the association between *CMTR2* deficiency and indisulam sensitivity, we used lung cancer organoid models[48] because tumor organoids retain the properties of the original tumors better than long-term cultured cell lines. The lung cancer organoid library contained a LADC organoid derived from a smoker that harbored a nonsense mutation (C > A transversion, AA change) in *CMTR2* alongside an oncogenic mutation in *EGFR* (Supplementary Table 8). This organoid was more sensitive to indisulam than *EGFR*-mutant organoids without *CMTR2* deficiency (Fig. 6b–d).

To confirm the link between *CMTR2* deficiency and increased drug sensitivity, we evaluated drug responses in *CMTR2*-knockout cells. These cells showed enhanced indisulam-induced growth inhibition and increased apoptosis (Fig. 6e–j). Additionally, in a xenograft mouse model generated by subcutaneously transplanting *CMTR2*-knockout A549 cells into immunodeficient mice, indisulam had favorable therapeutic effects (Fig. 6k–m). By contrast, overexpression of WT *CMTR2* in NCI-H1373 cells reduced drug sensitivity (Fig. 6n, o, and Supplementary Fig. 12a). Similar results were obtained with a different sulfonamide compound, E7820[49] (Supplementary Fig. 17a–g). Consistently, *CMTR2*-knockout A549 cells showed significantly enhanced sensitivity to two independent *RBM39*-targeting small interfering RNAs (siRNAs) compared with control cells (Supplementary Fig. 18a–d). Indisulam induced global splicing alterations in both A549 control and *CMTR2*-knockout cells (Supplementary Fig. 19a–d), and the number of differential splicing events was higher in *CMTR2*-knockout cell lines than in A549 control cells (Fig. 6p). This suggests that *CMTR2* deficiency sensitizes cells to sulfonamides by exacerbating splicing disruptions resulting from RBM39 degradation, similar to the effects observed in *SF3B1*-mutant cells treated with sulfonamides[47].

To further explore the vulnerability of *CMTR2*-deficient cells, we used gene effect scores from CRISPR knockout and RNA interference (RNAi) screens in the DepMap portal. After excluding genes with common vulnerabilities across all registered NSCLC cell lines, we identified 36 and 22 vulnerability genes from the CRISPR and RNAi

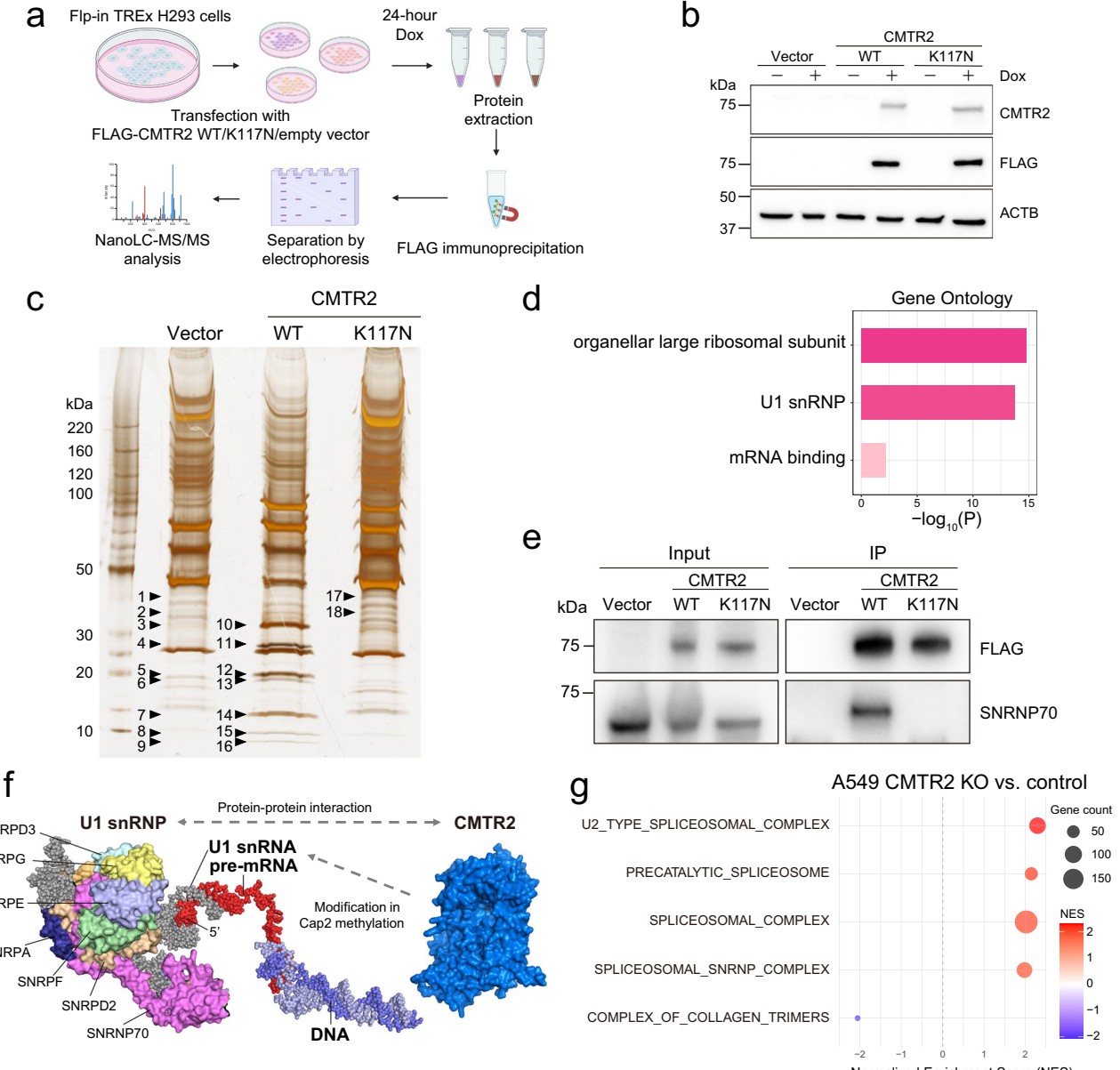

**Fig. 5 | Interactions between CMTR2 and spliceosomal components.**
**a** Schematic showing the experimental workflow for NanoLC-MS/MS analysis of FLAG-immunoprecipitated proteins following doxycycline (Dox)-induced expression of WT CMTR2 or K117N mutant proteins in Flp-in T-REx H293 cells. **b** Western blot analysis confirming Dox-induced expression of CMTR2 and FLAG proteins (representative of three independent experiments). **c** Silver-stained SDS-PAGE gel of FLAG immunoprecipitates indicating the bands excised for NanoLC-MS/MS analysis. Numbered bands correspond to background (1–9), WT CMTR2-specific (10–16), and K117N mutant-specific (17–18) proteins (data from a single experiment). **d** Gene Ontology (GO) enrichment analysis of proteins specifically enriched

in FLAG-WT CMTR2 immunoprecipitates. **e** Western blot analysis of SNRNP70 co-immunoprecipitation with FLAG-WT CMTR2 or K117N mutant protein (representative of two independent experiments). **f** Schematic showing the potential interaction between CMTR2, snRNP, and pre-mRNA based on the predicted CMTR2 protein structure (UniProt ID: Q8IYT2) from the AlphaFold Database and the complex structure of snRNP (PDB ID: 7BOY). **g** Bubble plot showing GSEA of transcriptome data from *CMTR2*-knockout clones compared with control A549 cells. GO cellular component gene sets that achieved statistical significance (FDR < 0.05) are shown. Figure panel **a** was created in BioRender (Nukaga, S. (2025) https://BioRender.com/jqfhjfm).

screens, respectively (Supplementary Fig. 20a, b). Enrichment analysis of these vulnerability genes revealed significant enrichment in mRNA splicing pathways and spliceosomal complex-related gene sets (Supplementary Fig. 20c, d). Analysis of the STRING biological interaction network[44] further revealed interactions between RBM39 and the identified vulnerability genes in the mRNA splicing gene set (Supplementary Fig. 20e). Taken together, these findings suggest that *CMTR2*-deficient cells show increased vulnerability in relation to their splicing machinery. Sulfonamides exert antitumor effects by targeting this vulnerability through degradation of RBM39.

### CMTR2 deficiency enhances responses to ICB therapy
Disruptions in RNA splicing in cancer cells can generate tumor-specific neoantigens, which may elicit immune responses[16]. For example, uveal melanoma, which harbors RNA splicing disruptions due to mutations in *SF3B1*, generates splicing-derived neoantigens[50]. It is thus plausible that *CMTR2*-deficient tumors, which exhibit extensive AS alterations, could respond favorably to ICB therapy. To test this hypothesis, we utilized a syngeneic mouse model in which LLC cells were transplanted into C57BL/6 mice. Both *CMTR2*-knockout and WT LLC cells were subcutaneously implanted, and the therapeutic effects of indisulam

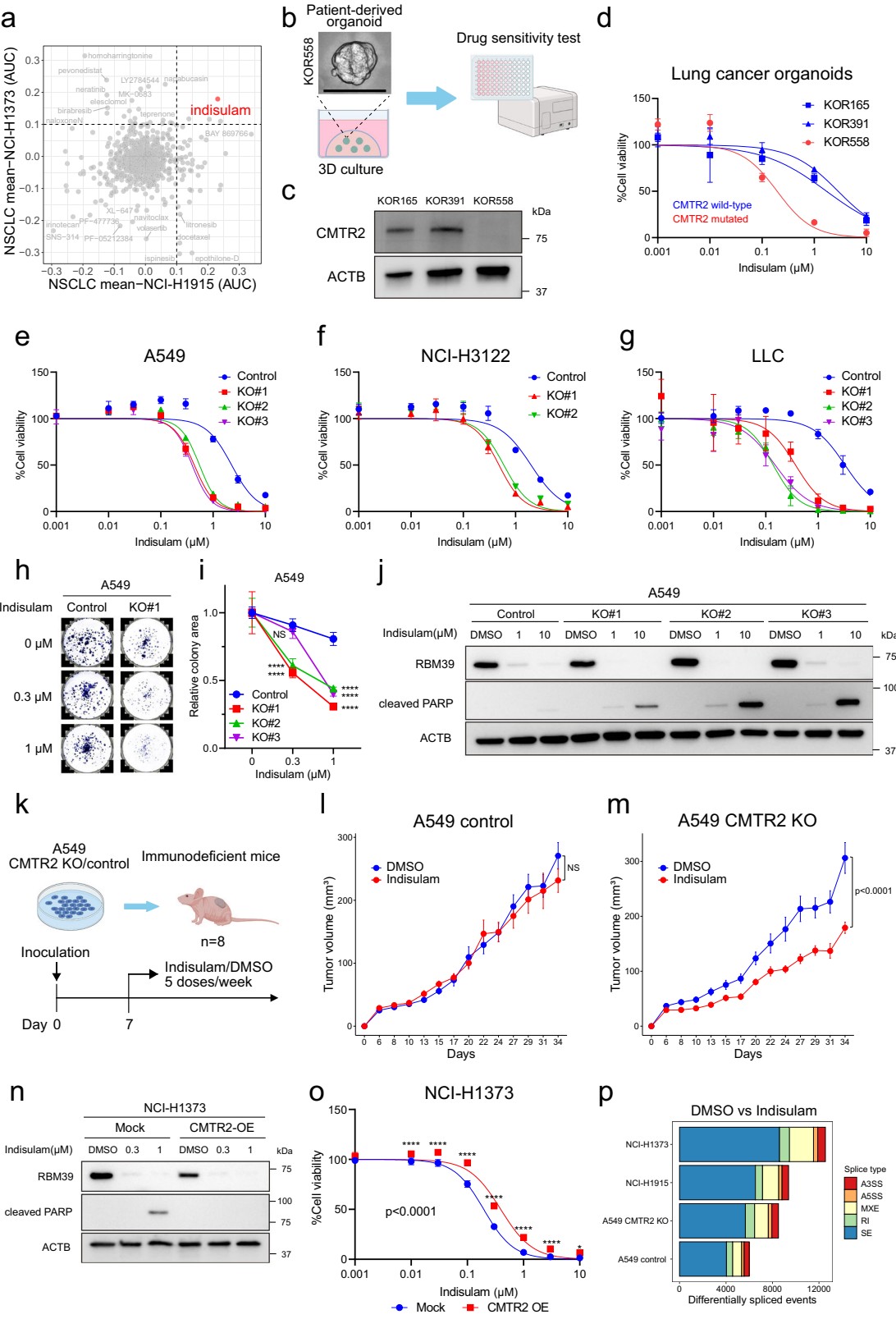

and/or anti-PD-1 antibody treatment were evaluated (Fig. 7a). Treatment with the anti-PD-1 antibody significantly suppressed growth of *CMTR2*-knockout LLC cells but had no effect on parental LLC cells, consistent with the reported resistance of LLC cells to anti-PD-1 antibody monotherapy[51] (Fig. 7b–e). *CMTR2*-knockout LLC cells also demonstrated increased sensitivity to indisulam, further supporting

the role of *CMTR2* deficiency in sulfonamide responsiveness observed in our previous experiments (Fig. 6k–m).

In both the NCC and TCGA cohorts, LADC cases harboring *RBM10*, *U2AF1*, or *CMTR2* mutations exhibited numerous splicing abnormalities, and the frequency of abnormal splicing events was higher in *CMTR2*-mutant tumors than in other spliceosomal mutant tumors

**Fig. 6 | Therapeutic vulnerability of *CMTR2*-deficient cells to the splicing modulator indisulam. a** Scatterplot showing drug sensitivity according to AUC differences (NSCLC cell lines: NCI-H1373 or NCI-H1915). **b** Schematic of drug sensitivity testing using patient-derived organoids (scale bar, 100 μm). **c** Western blot analysis of CMTR2 expression in lung cancer organoids. **d** Drug-response curves of *CMTR2* mutant (KOR558) and WT (KOR165 and KOR391) lung cancer organoids after 7 days of treatment with indisulam (*n* = 4 technical replicates; mean ± SD). **e**–**g** Drug-response curves of CRISPR-mediated *CMTR2*-knockout clones and control cells [A549 (**e**), NCI-H3122 (**f**), and LLC (**g**)] after 5 days of treatment with indisulam (*n* = 4 technical replicates; mean ± SD). **h, i** Colony formation assay using CRISPR-mediated *CMTR2*-knockout and control A549 cells treated with indisulam for 11 days. Representative images (**h**) and quantification of the relative colony area for all clones (**i**) are shown. **j** Western blot analysis of CRISPR-mediated *CMTR2*-knockout clones and control cells (A549) after treatment with increasing doses of indisulam for 72 h (representative of three independent experiments). **k** Schematic showing the drug treatment and inoculation schedules. **l, m** Charts showing tumor volume in BALB/c-nude mice bearing control (**l**) or CRISPR-mediated *CMTR2*-knockout (**m**) A549 cells treated with indisulam. Day 0 represents the time of tumor cell implantation. *n* = 8/group. NS: not significant, two-way repeated measures ANOVA. Data are presented as the mean ± SEM. **n** Western blot analysis of NCI-H1373 cells transduced with *CMTR2* (CMTR2-OE) or mock control and treated for 72 h with increasing doses of indisulam. **o** Drug-response curves of CMTR2-OE or mock control cells treated for 5 days with indisulam. **p** Horizontal bar chart showing the number of significant differentially spliced events in NCI-H1373, NCI-H1915, and A549 *CMTR2*-knockout and control cells treated with indisulam (FDR < 0.05 and |ΔPSI| ≥ 0.1). For **i** and **o**: *n* = 4 technical replicates, mean ± SD. The *P*-value was calculated using a two-way ANOVA. Asterisks indicate significant differences between the control and experimental groups at each concentration (Bonferroni's multiple comparisons test; ****p < 0.0001, * p = 0.013). Figure panels **b** and **k** were created in BioRender (Nukaga, S. (2025) https://BioRender.com/179bqky, https://BioRender.com/. hmgp2kd).

(Fig. 3b and Supplementary Fig. 6b). Based on these observations, we retrospectively analyzed the efficacy of ICB therapy in NSCLC patients with truncating *CMTR2* mutations. In three datasets that included both ICB therapy response data and somatic mutation profiles derived from whole-exome sequencing[52–54], we identified 11 non-squamous NSCLC cases with truncating *CMTR2* mutations. The disease control rate was 90.9% (10/11) in patients with truncating *CMTR2* mutations compared with 68% (237/350) in those without. Durable clinical benefit was achieved in 64% (7/11) of patients with truncating *CMTR2* mutations compared with 46% (171/373) of those without. Median progression-free survival (PFS) was 9.2 months for patients with truncating *CMTR2* mutations compared with 5.1 months for those without (Fig. 7f). While the difference of PFS was not statistically significant by the log-rank test due to the limited sample size (*p* = 0.471), the consistent trend toward improved clinical outcomes suggests that *CMTR2* deficiency is associated with increased responsiveness to ICB therapy. Additionally, we identified an institutional case harboring a truncating *CMTR2* mutation with available ICB treatment information, in whom there was a clinical response to pembrolizumab (an anti-PD-1 antibody) treatment (Supplementary Fig. 21 and Supplementary Data 3).

## Discussion

In this study, we identified *CMTR2* as a therapeutic target gene for LADC by integrating RNA splicing metrics and mutation data from our Japanese cohort and the TCGA LADC cohort. In a previous report, loss-of-function mutations in *CMTR2* were observed in LADC[11]. Supporting this observation, mutations in *CMTR2* were detected in 3.8% (48/1270) of LADC cases, and the majority were truncating mutations, which formed an isolated cluster within the RNA splicing landscape. The CMTR2 protein plays a crucial role in Cap2 2′-O-ribose methylation of the 5′-cap structure of RNA, a key post-transcriptional modification. *CMTR2* mutations caused significant alterations in RNA splicing, thereby contributing to the formation of LADC with distinct molecular subtypes. This RNA splicing landscape offers an approach to understanding tumor heterogeneity and identifying potential therapeutic targets. This study provides evidence of a biological link between *CMTR2* mutations and RNA splicing abnormalities.

The involvement of *CMTR2* in carcinogenesis was suggested in previous studies. For instance, loss of *CMTR2* promotes proliferation of KRAS-driven lung cancer cells[55]. Germline loss-of-function variants of *CMTR2* are significantly associated with an increased risk of LADC and cutaneous melanoma[56]. Additionally, the TRACERx study, a comprehensive multi-sampling analysis of lung cancer, identified *CMTR2* as a target of truncal mutations[57]. Consistent with these findings, the results of this study indicate that most *CMTR2* mutations are truncal, supporting the hypothesis that *CMTR2* functions as a tumor suppressor gene during LADC development. To investigate how *CMTR2* deficiency contributes to tumor formation, we conducted differential gene expression and pathway enrichment analyses using clinical specimens and cell models. While *CMTR2*-knockout cells exhibited significant changes in expression of RNA splicing-related cellular components, gene sets related to molecular functions or biological processes were not significantly enriched. These results highlight the challenges of understanding how *CMTR2* deficiency affects carcinogenesis through gene expression data and suggest that alternative investigative approaches are needed.

We also observed that *CMTR2* mutations were more common in smokers than in non-smokers. Previous reports indicate that stop-gain mutations are enriched in a single base substitution signature associated with smoking, specifically affecting 34 tumor suppressor genes involved in cancer hallmark pathways[35]. *CMTR2* was identified as one of these 34 genes. Furthermore, our analysis of TCGA PanCancer Atlas data revealed that LADC is the cancer type with the highest frequency of truncating *CMTR2* mutations, further supporting the notion that *CMTR2* is a target of smoking-related mutagenesis in LADC.

Our immunoprecipitation experiments and protein–protein interaction database[45] analysis showed that the CMTR2 protein physically interacts with multiple components of snRNPs, which comprise snRNAs and associated proteins. U1, U2, U4, and U5 snRNAs are fully or nearly fully Cap2 methylated in different cell types[58]. Knockout of *CMTR2* in HEK293T cells results in loss of Cap2 methylation on U1 snRNA[27], whereas Cap2 modification of U2 snRNA is essential for the formation of spliceosome complexes and RNA splicing[59]. Therefore, the absence of Cap2 modification in snRNAs due to *CMTR2* deficiency may destabilize the spliceosome complex and contribute to RNA splicing anomalies. *CMTR2* knockout altered the expression of a broad range of splicing-related genes beyond snRNPs, indicating that *CMTR2* orchestrates the global spliceosomal network. These findings uncover a previously unrecognized RNA splicing deficiency in human cancers.

Our splicing analysis, conducted primarily through short-read RNA-seq, revealed characteristic splicing abnormalities associated with *CMTR2* deficiency; however, short-read-based tools have inherent limitations, including potential false positives and false negatives, with respect to resolving complex splicing events such as trans-splicing, as well as quantifying accurate isoforms. Although we validated key splicing events in *CMTR2*-knockout cells using long-read sequencing, which showed strong concordance with the short-read data obtained from *SPPL2A* and *SMARCA1*, comprehensive genome-wide analysis of the long-read sequencing data across multiple samples was not feasible. Future systematic long-read sequencing analyses would allow more comprehensive characterization of splicing dysregulation and complexity driven by *CMTR2*-deficiency.

Targeting RNA splicing has emerged as a promising strategy for cancer therapy. We demonstrated that *CMTR2*-deficient cancer cells exhibit vulnerabilities in the mRNA splicing machinery and are sensitive to RBM39 degraders, such as sulfonamides. This aligns with

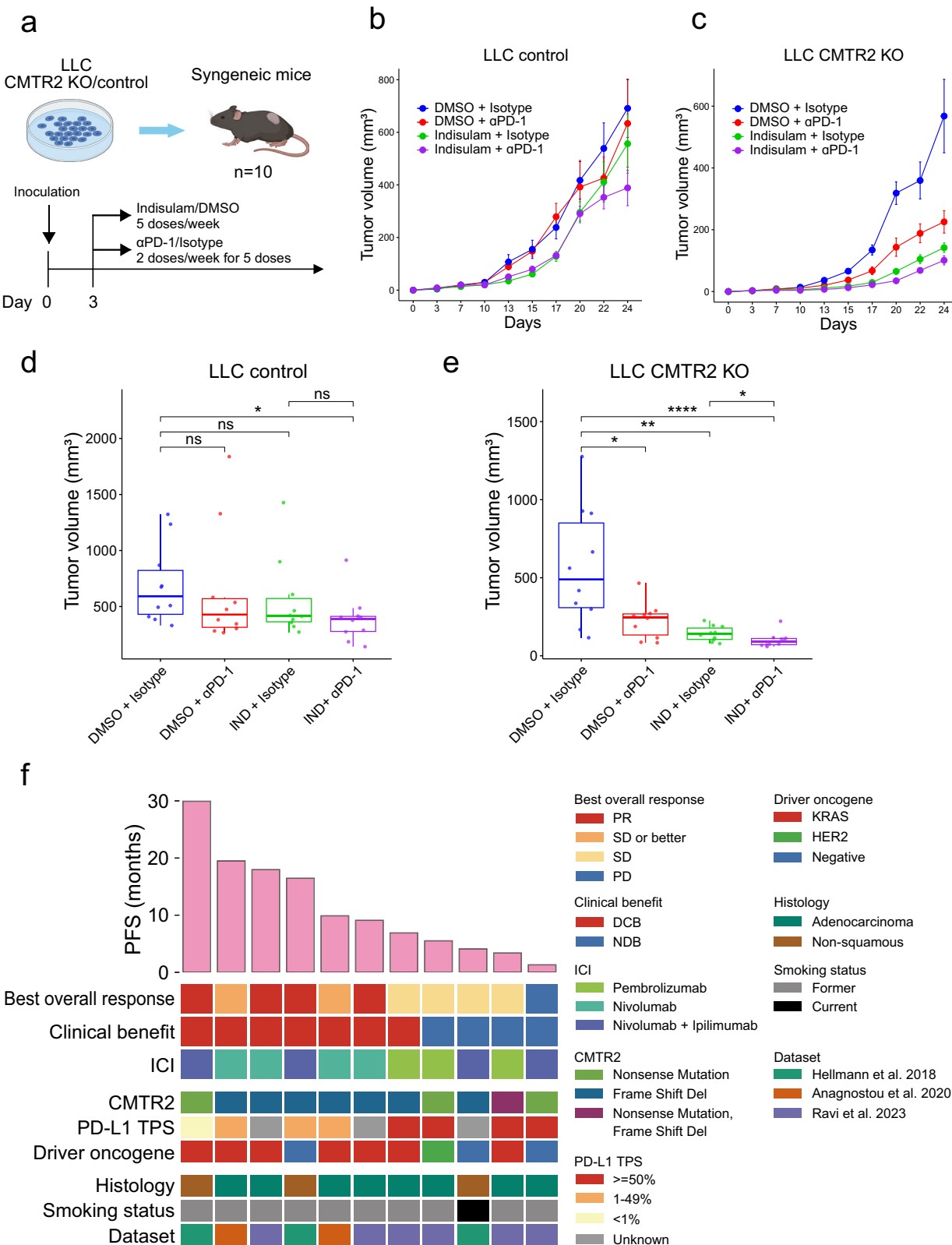

previous findings showing that cancer cells harboring spliceosomal gene mutations are more susceptible to pharmacological splicing perturbations[47]. Although several Phase I and II clinical trials of sulfonamides have been conducted, only mild-to-moderate anticancer effects were observed, which is likely due to the limited understanding of their pharmacologic mechanisms as RBM39 degraders and the lack of predictive biomarkers[60]. Given their high tolerability, it may be

possible to demonstrate their effectiveness in clinical trials with appropriately selected patients. Sulfonamides exhibit antitumor effects through pleiotropic mechanisms[61], and mechanisms other than RBM39 degradation might underlie their effects on *CMTR2*-deficient cancers. Further investigation of the mechanisms of action of sulfonamides could lead to the identification of additional therapeutic targets for *CMTR2*-deficient cancers.

**Fig. 7 | Efficacy of ICB therapy in *CMTR2*-deficient tumors. a** Schematic showing the drug treatment and inoculation schedules. **b, c** Tumor volume in C57BL/6 mice bearing control (**b**) or CRISPR-mediated *CMTR2*-knockout (**c**) LLC cells treated with vehicle, indisulam, an anti-PD-1 antibody, or a combination of indisulam and an anti-PD-1 antibody (*n* = 10/group). Day 0 represents the time of tumor cell implantation. Data are presented as the mean ± SEM. **d** Boxplot showing the tumor volume from (**b**) on day 24. The center line indicates the median value, the lower and upper hinges represent the 25th and 75th percentiles, respectively, and the whiskers denote the 1.5× interquartile range. Statistical analysis was performed using the two-sided Wilcoxon rank-sum test: DMSO + isotype *versus* indisulam + anti-PD-1, *\*p* = 0.019. **e** As in (**d**) but using data from (**c**). DMSO + isotype *versus* DMSO + anti-

PD-1, *\*p* = 0.011; indisulam + isotype, *\*\*p* = 0.0011; indisulam + anti-PD-1, *\*\*\*\* p* = 7.6e-05; indisulam + isotype *versus* indisulam + anti-PD-1, *\*p* = 0.035. **f** Efficacy of ICB therapy in non-squamous NSCLC cases harboring a *CMTR2* truncating mutation. Datasets were obtained from Hellman et al.[52], Anagnostou et al.[53], and Ravi et al.[54]. Top: PFS of cases treated with an immune checkpoint inhibitor (ICI). PR partial response, SD stable disease, PD progressive disease, DCB durable clinical benefit, NDB no durable benefit, PD-L1 programmed death-ligand 1, TPS, tumor proportion score. DCB was defined as SD or PR lasting longer than 6 months; all other cases were considered to have NDB. Figure panel **a** was created in BioRender (https://BioRender.com/. w53g680).

In this study, *CMTR2* knockout increased sensitivity to ICB therapy in LLC cells, which are typically resistant to anti-PD-1 treatment. *CMTR2*-mutant tumors exhibited a higher number of both known (annotated) and novel (unannotated) splicing events. The novel splicing events generate tumor-specific RNA isoforms, which suggests that *CMTR2* deficiency can result in the production of splicing-derived neoantigens. This is supported by the results of a retrospective analysis indicating that patients with *CMTR2*-mutant lung cancers respond favorably to ICB therapy. *CMTR2* mutations could thus serve as a biomarker for identifying patients who may benefit from ICB therapy. AT1965, an inhibitor of *CMTR2* that is currently in clinical trials (NCT06234098), activates B cell-mediated immune responses in a lung cancer model and shows synergistic antitumor effects when combined with PD-1 inhibition[62]. This finding further supports the notion that *CMTR2* deficiency enhances the efficacy of ICB therapy. However, as a limitation of this study, the identification of neoantigens generated by *CMTR2* mutations and the subsequent activation of immune responses were not experimentally validated. It is also possible that *CMTR2* deficiency enhances the efficacy of ICB therapy through other mechanisms, such as aberrant splicing of immune-related genes. Additionally, we could not directly assess Cap2 methylation changes in our *CMTR2*-knockout models. Detailed investigation of these Cap2 alterations and their functional consequences could reveal how *CMTR2* influences tumor progression and immune responses.

Overall, *CMTR2*-deficient tumors present a therapeutic opportunity through splicing perturbation. While our preliminary data suggest that such tumors may exhibit enhanced responses to ICB therapy, larger prospective studies and mechanistic investigations are needed to validate this and to determine its clinical significance.

## Methods
### Patients
The NCC cohort of 1017 cases was diagnosed with lung cancer at the NCC Hospital in Tokyo, Japan, between 2011 and 2017. The cohort comprised 568 males and 449 females. The median age was 66 years (range: 26–89 years; males: median 65 years, range 26–89; females: median 66 years, range 35–88). Sex was self-reported during patient enrollment, and confirmed by review of medical records. Gender identity was not assessed specifically in this study. Pathological diagnosis was performed according to the 7th edition of the World Health Organization classification. Non-physician healthcare professionals used questionnaires to collect information about each patient's exposure to smoking. Never-smokers were defined as individuals who had never smoked or had smoked fewer than 100 cigarettes in their lifetime. Fresh frozen cancerous and non-cancerous lung tissues obtained through surgical resection were available from the NCC Biobank and used for molecular analyses. Tissue samples were consumed during the analytical procedures. Residual samples, which are available for some cases, are held at the NCC Biobank and can be shared with other researchers under a collaborative research framework, subject to approval from the NCC Biobank and the Institutional Review Board of the NCC. Contact the corresponding authors for sharing requests. No sex or gender analysis was carried out in this

study because the research questions focused on molecular mechanisms of lung cancer that are not thought to differ according to sex or gender. Participants received no financial compensation.

### RNA-seq
Total RNA was extracted from frozen normal lung tissues or tumor samples from the NCC cohort using an AllPrep DNA/RNA Mini Kit (QIAGEN). In addition, total RNA was extracted from cell lines using an RNA RNeasy Mini Kit (QIAGEN).

**Short-read sequencing.** Poly(A)-selected strand-specific libraries were prepared using a TruSeq Stranded mRNA library prep kit (Illumina). Sequencing was performed using the Illumina NovaSeq6000 platform, with 150-bp paired-end reads. For TCGA transcriptome data, sequencing data were obtained from the Genomic Data Commons (GDC). Alignment to hg38 (reference genome) was performed using STAR ver. 2.7.9a[63]. Gene fusions were estimated using STAR-Fusion ver. 1.11.0[64]. Read counts and expression levels, normalized to transcripts per kilobase million, were estimated using StringTie ver. 2.0.4[65] or RSEM ver. 1.3.3[66].

**Long-read direct RNA-seq.** Sequencing libraries were prepared using a SQK-RNA004 kit (Oxford Nanopore Technologies) according to the manufacturer's instructions. The prepared libraries were sequenced on a PromethION 2 Solo instrument (Oxford Nanopore Technologies) over 72 h. Base calling was performed using Dorado basecaller ver. 0.9.1 + c8c2c9f[67]. Low-quality reads (average Q < 7) were filtered using chopper[68]. Alignment to hg38 was performed using minimap2 ver. 2.28-r1209[69]. Transcript annotation, novel isoform discovery, and expression quantification were performed using Bambu ver. 3.8.3[70] with GENCODE release 41 comprehensive annotations as the reference.

### AS analysis
Detection of AS events and calculation of PSI values, also known as the exon inclusion ratio, were performed using rMATS turbo ver. 4.1.2. The detailed types of AS events that can be detected using rMATS were described in the original paper[14]. rMATS in a non-standard mode, which does not require two conditions, was used to calculate PSI values for t-SNE and PCA. All of the BAM files to be processed were listed in the --b1 file, omitting the --b2 file, and used the optional argument --statoff. The default statistical comparison mode was used to detect differentially spliced events in cell lines and human samples. For dimensionality reduction analyses (t-SNE and PCA), the following filters were applied to PSI values calculated by rMATS, and only significant AS events that passed these filters were used: average PSI values between 0.05 and 0.95 to exclude constitutive splicing events, average sum read count (inclusion count and skipping count) ≥10 to ensure sufficient coverage, percentage of samples with a missing PSI value < 0.05 to minimize the impact of missing data, and delta PSI values (maximum PSI values − minimum PSI values) >0.3 to focus on events with high biological variability. These stringent criteria ensured robust and unbiased dimensionality reduction. t-SNE was performed

using the R package Rtsne ver. 0.16. PCA was performed using the prcomp function in R ver. 4.2.2. Sequence logs were created with ggseqlogo R package ver. 0.2[71] to visualize nucleotide frequency and conservation at splice sites.

### Use of public databases for AS analysis of the *SPPL2A* and *SMARCA1* genes

PSI data for normal tissues and cell lines were obtained from VastDB[72] (http://vastdb.crg.eu/). For TCGA cohort analysis, PSI data from the SplAdder project[16], obtained from the OncoSplicing database[73], were used. The chromosomal position of the *SPPL2A* and *SMARCA1* SE events in each database corresponds to the event detected in the present study, as shown in the source data of Supplementary Fig. 13.

### Differential gene expression and gene enrichment analysis

Differential gene expression analysis was performed using R DESeq2 package ver. 1.36.0[74]. Genes with an adjusted $P$-value < 0.05 and a |log2-fold change | >1 were considered to be differentially expressed genes (DEGs). DEGs were analyzed using Metascape[75] (http://metascape.org) with default parameters for tissue/cell-specific gene signature enrichment analysis based on PaGenBase[76]. GSEA was performed using the GSEA module in GenePattern[77]. The analysis utilized the MSigDB GO Cellular Component ontology gene sets. The permutation type was set to 'gene_set' with 1000 permutations. Core enrichment genes from GSEA were annotated with their spliceosomal complex, class, or family associations using the Spliceosome Database[78].

### EMT score calculation

An established method using the two-sample Kolmogorov-Smirnov test based on EMT signature gene expression levels[15,20] was employed to score the EMT status of NCC samples. The scoring script and signature genes were retrieved from https://github.com/Xinglab/rmats-turbo-tutorial. Samples were classified as mesenchymal (statistic EMT score > 0 with FDR < 0.05), epithelial (statistic EMT score <0 with FDR < 0.05), or intermediate (all other samples).

### Neuroendocrine score calculation

Neuroendocrine scores were calculated by single-sample GSEA using GSVA R package ver. 3.2.0[79] as previously described[22]. The neuroendocrine gene signature was obtained from a previous report[23].

### Whole-exome sequencing

**DNA extraction and sequencing.** DNA was extracted from tumor and matched normal tissues (e.g., blood or normal lung tissue) using a QIAamp DNA Mini Kit or AllPrep DNA/RNA Mini Kit (QIAGEN). For a small subset of samples for which RNA-seq data were available but DNA of sufficient quality could not be obtained, whole-exome sequencing was not performed. Exon capture was performed using SureSelect Human All Exon exome capture kits (Agilent Technologies). Tumor and matched normal tissues were sequenced on the HiSeq2500 or NovaSeq6000 platform (Illumina) using 2 × 75 bp/2 × 100 bp or 2 × 150 bp paired-end reads, respectively. The sequencing depth was set to 150× for tumor samples and 100× for normal controls.

**Data analysis.** FASTQ data were subjected to genome mapping and variant calling using an in-house data analysis pipeline. Alignment to the reference genome (hg38) was performed using Parabricks ver. 3.1.3 (Nvidia), which implements algorithms equivalent to BWA ver. 0.7.15[80] and GATK ver. 4.1.0[81]. Duplicate reads were identified and marked, followed by realignment and recalibration of base quality scores. The mapped data were then exported in BAM format.

**Variant calling.** Somatic single-nucleotide variants (SNVs) were called using Mutect2 (GATK ver. 4.1.2.0) in tumor-normal mode[82]. Variants were filtered using FilterMutectCalls and FilterAlignmentArtifacts, and only those with PASS in the FILTER field were selected. Additionally, a panel of "normals" derived from blood was used to filter background mutations. Small insertions and deletions (InDels) were called using both Mutect2 and Strelka2[83], with only PASS-filtered variants selected.

**Variant annotations.** Detected variants were annotated using OncoKB[84]. Oncogenic variants were defined as those annotated as "oncogenic" or "likely oncogenic" in the OncoKB database. Oncogenic SNVs were included if called by Mutect2, whereas InDel variants were included if called by Mutect2 or Strelka2. Driver genes analyzed in Fig. 1c were validated using the Integrative Genomics Viewer[85]. For *U2AF1* mutations, there are issues with the GRCh38 (hg38) reference build that prevent detection of these mutations[86] and therefore variant calling was performed using Mutect2, with the hg19 genome as the reference.

### TCGA genome and acquisition of clinical data

Genomic mutations, the tumor mutation burden, the MSIsensor score, and copy number alteration data for TCGA cohorts were obtained from cBioPortal (https://www.cbioportal.org/). For the TCGA LADC cohort, information about smoking history was retrieved using the R package TCGAmutations ver. 0.3.0.

### Computational identification of driver mutations

To identify significantly mutated genes in the TCGA LADC cohort, a consensus informatics approach was performed using MutPanning ver. 2.0[32], OncodriveFML ver. 2.1.3[33], DriverPower ver. 1.0.2[87], and OncodriveCLUSTL ver. 1.1.3[88]. The details of each algorithm are described in the original papers. Briefly, MutPanning and OncodriveCLUSTL are designed to detect cancer driver genes by modeling the mutation probability of each genomic position according to its neighboring nucleotide context and background mutation rate. OncodriveFML and DriverPower are designed to detect cancer driver mutations by combining functional scoring schemes with a background mutation model. The input MAF files were obtained from the GDC data portal (https://portal.gdc.cancer.gov), and each analysis was performed using default settings.

### Analysis of intratumor heterogeneity

Intratumor heterogeneity in samples with *CMTR2* SNV mutations from the CMTR2 cluster within the NCC cohort was analyzed using PyClone-VI ver. 0.1.1[34] with default parameters. This analysis utilized SNVs called by Mutect2 (GATK ver. 4.1.2.0)[82] from whole-exome sequencing data, along with copy number variant profiles and purity estimates calculated using Facets ver. 0.6.2[89]. Samples lacking determinable purity were excluded. To visualize subclonal evolutionary relationships, phylogenetic trees were constructed using ClonEvol ver. 0.99.11[90].

### Heat-map and hierarchical clustering

To visualize the diversity of exon-skipping events in the clusters identified in the NCC cohort, a heat-map with unsupervised hierarchical clustering (Spearman's correlation distance and Ward clustering) was drawn using exon-skipping events detected by rMATS analysis. Analysis was performed using ComplexHeatmap ver. 2.12.1 in the R package[91].

### Prediction of CMTR2 protein structure and mutation-induced pathogenicity

The predicted structure of the CMTR2 protein (UniProt ID: Q8IYT2; CMTR2_HUMAN) was obtained from the AlphaFold Protein Structure Database[92] (https://alphafold.ebi.ac.uk/). This structure was aligned with the theoretical model of the catalytic domain of CMTR2[37] to display the 5′ cap (m7GpppGGAA) on RNA and SAM. AM pathogenicity scores for all possible missense mutations in *CMTR2* were obtained from the AM dataset[39]. Mutations were classified according to

previously reported cutoff values: likely pathogenic (score >0.564), likely benign (score <0.34), or ambiguous (0.34≤ score ≤0.564). These scores were used to color-code the protein structure and visualize the distribution of scores across all possible missense mutations.

## Analysis of publicly available gene dependency and drug sensitivity data

Gene effect (dependency) scores, which were evaluated by RNAi and CRISPR knockout, were obtained from the DepMap portal (https://depmap.org/portal/). As defined by DepMap, a score of 0 defines a gene that is not essential, whereas a score of −1 corresponds to the median of all common essential genes. CRISPR screening (DepMap Public 22Q+Score, Chronos) and RNAi screening (Achilles+DRIVE +Marcotte, DEMETER2) data were used for gene dependency analysis. AUC values indicating drug sensitivity were calculated from the PRISM Repurposing Secondary Screen[46].

## Cell lines

A549 (CCL-185), NCI-H1915 (CRL-5904), NCI-H1373 (CRL-5866) and LLC (CRL-1642) cells were purchased from the American Type Culture Collection. NCI-H3122 was provided by Dr William Pao (Vanderbilt University, Nashville, TN, USA). Flp-in T-REx 293 cells (R78007) were purchased from Thermo Fisher Scientific. A549 (male), NCI-H3122 (male), NCI-H1915 (female), NCI-H1373 (male), and Flp-in T-REx 293 (female) are human-derived cell lines. LLC cells are mouse-derived. A549, NCI-H3122, NCI-H1915, and NCI-H1373 cells were cultured in RPMI-1640 medium (FUJIFILM Wako Pure Chemical), whereas LLC and Flp-in T-REx 293 cells were cultured in DMEM (FUJIFILM Wako Pure Chemical). All media were supplemented with 10% fetal bovine serum (Thermo Fisher Scientific). Cultures were maintained at 37 °C in a humidified 5% $CO_2$ incubator.

## Patient-derived lung cancer organoids

*EGFR* mutation-positive organoids with and without *CMTR2* mutations were used for experiments; the organoids were retrieved from the in-house lung cancer organoid library[48]. Clinical samples used to establish organoids were obtained from patients at Keio University Hospital or Kawasaki Municipal Hospital. Organoid culture was performed based on a previous report[48]. Organoids were embedded in Matrigel (Corning) and cultured in Advanced DMEM/F12 supplemented with penicillin/streptomycin, 10 mM HEPES, 2 mM GlutaMAX, 1× B27 (Thermo Fisher Scientific), 10 nM gastrin I (Sigma-Aldrich), and 1 mM N-acetyl-L-cysteine (Sigma-Aldrich). For organoid maintenance, the medium was supplemented with the following factors and inhibitors: 50 ng/mL murine recombinant EGF (Thermo Fisher Scientific), 100 ng/mL human recombinant IGF-1 (BioLegend), 50 ng/mL human recombinant FGF-2 (PeproTech), 1 μg/mL human recombinant R-spondin1 (R&D Systems), 5% Afamin-Wnt-3A serum-free conditioned medium[93], 100 ng/mL murine recombinant noggin (PeproTech), 500 nM A83-01 (Tocris Bioscience), and 10 μM Y-27632 (FUJIFILM Wako Pure Chemical). The seeded organoids were maintained in an incubator with an atmosphere of 5% $CO_2$ and 20% $O_2$.

## Reagents

Indisulam and E7820 were purchased from Selleck Chemicals. Anti-mouse PD-1 (Ultra-LEAF Purified anti-mouse CD279 (PD-1); catalog No. 114122; clone RMP1-14; lot B434686) and isotype control (Ultra-LEAF Purified Rat IgG2a, κ Isotype Ctrl; catalog No. 400574; clone RTK2758; lot B449464) antibodies were purchased from BioLegend and used for in vivo treatments. The following primary antibodies were used for immunoblot analysis: anti-β-actin (1:1000; catalog No. 3700, clone 8H10D10; lot 13, Cell Signaling Technology); anti-cleaved PARP (1:1000; catalog No. 5625, clone D64E10; lot 13, Cell Signaling Technology); anti-RBM39 (1:1000; catalog No. HPA001591, polyclonal; lot 000044370, Sigma-Aldrich); anti-FLAG (1:3000; catalog No. F1804,

clone M2; lot 0000278731, Sigma-Aldrich); anti-CMTR2 (1:500; catalog No. PA5-61696, polyclonal; lot YG3983628A, Thermo Fisher Scientific); and anti-SNRNP70 (1:1000; catalog No. SC390899, clone C-3; lot G2122, Santa Cruz Biotechnology).

## Establishment of the NCI-H1373 CMTR2-overexpressing stable cell line

Control cells and stable NCI-H1373 cells overexpressing CMTR2 were generated by lentiviral infection of the pLV[Exp]-EGFP:T2A:Puro-EF1A > hCMTR2[NM_018348.6]/FLAG or pLV[Exp]-EGFP:T2A:Puro-EF1A > ORF_Stuffer construct (custom-synthesized by VectorBuilder). Lentiviruses expressing CMTR2 were produced by transfecting the expression plasmid into 293FT cells along with ViraPower Packaging Mix (Thermo Fisher Scientific) using Lipofectamine 3000 (Thermo Fisher Scientific). For stable expression, NCI-H1373 cells (70% confluent) were infected with empty or CMTR2-expressing lentiviruses and then treated with puromycin (2 μg/mL) for 2 weeks. Mass-cultured puromycin-resistant cells were used for the assays.

## Establishment of *CMTR2*-knockout cell lines using CRISPR-Cas9

To create *CMTR2*-knockout cell lines, CRISPR-Cas9-mediated gene knockout was performed as described in a previous report[94] and in accordance with the manufacturer's instructions, but with some modifications. Single guide RNAs (sgRNAs) were designed using the Benchling CRISPR Guide RNA Design Tool (https://benchling.com/). CRISPR RNAs (crRNAs; Alt-R CRISPR-Cas9 custom crRNA) and trans-activating crRNAs (tracrRNAs; Alt-R CRISPR-Cas9 tracrRNA, catalog No. 1072532) were obtained from Integrated DNA Technologies. sgRNAs (150 pmol) were formed by hybridizing crRNAs with tracrR-NAs, and the ribonucleoprotein complex was formed by addition of 122 pmol Cas9 nuclease (Integrated DNA Technologies) in vitro. Alt-R CRISPR-Cas9 Negative Control crRNA #1 (Integrated DNA Technologies, catalog No. 1079138) was used for control experiments. The reaction mixtures were nucleofected into A549, NCI-H3122, or LLC cells ($1 \times 10^5$ cells) suspended in 20 μL of Nucleofector solution containing a supplement (Lonza) using a Lonza 4D-Nucleofector (Lonza) in CM-130 mode. To confirm gene knockout, DNA was extracted from single clones using QuickExtract™ DNA Extraction Solution (Lucigen). Induced frameshift mutations were confirmed by Sanger sequencing (FASMAC), and the knockout efficiency was analyzed using the Synthego ICE CRISPR Analysis Tool (https://ice.synthego.com). The following sgRNA target sequences were used:

Human *CMTR2* gene-specific sgRNA: CCCATCGGTTTCCTTGTC AT; mouse *Cmtr2* gene-specific sgRNA: GATTACGGATGACCGGCTGA.

## Cell proliferation assay

Cell lines and organoids were used for cell proliferation assays. For cell lines, cells were seeded at previously optimized densities in 96-well white-walled tissue culture-treated plates. After 24 h, serially diluted inhibitors were added to the wells. For organoids, 1500 cells in 8 μL of Matrigel were seeded per well and immediately treated with 50 μL of serially diluted inhibitors. In both cases, control wells were treated with the same concentration of vehicle [dimethyl sulfoxide (DMSO)]. Cell viability was measured using CellTiter-Glo luminescent reagent (Promega) with EnVision (Perkin Elmer) or Glomax Discover (Promega). Measurements were taken 5 and 7 days after treatment of cell lines and organoids with inhibitors. Data were displayed graphically using GraphPad Prism ver. 9.5.1 (GraphPad Software Inc.).

## Colony formation assay

Cells were seeded into a 6-well plate at a previously optimized density, cultured for 24 h, and then treated with the indicated concentrations of inhibitors. Every 3 days, the medium was replaced with fresh medium containing inhibitors. After 11 or 14 days of treatment, colonies were fixed with methanol and stained with Coomassie Brilliant Blue

solution. Image tiling and stitching of stained cells were performed using the EVOS™ M7000 Imaging System (Thermo Fisher Scientific). Colony area was quantified using Celleste™ 6 Image Analysis Software (Thermo Fisher Scientific).

### siRNA-mediated knockdown
Cells ($1.0 \times 10^5$/well) were seeded in 6-well plates and transfected 24 h later with siRNA oligonucleotides at the indicated concentrations using Lipofectamine RNAiMAX (Thermo Fisher Scientific). Silencer Select predesigned siRNA (Thermo Fisher Scientific, ID: s18416) for siRNA#1 and MISSION Predesigned siRNA (Sigma-Aldrich, ID: SASI_Hs02_00338551) for siRNA#2 were used. Silencer Select Negative Control No. 1 siRNA (Thermo Fisher Scientific, catalog No. 4390843) was used as a negative control at a final concentration of 5 nM. At 48 h post-transfection, cells were harvested for protein analysis or reseeded into 96-well plates at a density of 500 cells per well for cell viability assays. Cell viability was measured 6 days after transfection using CellTiter-Glo 2.0 luminescent reagent (Promega) and an EnVision plate reader (PerkinElmer).

### Immunoblot analysis
Cells were lysed in RIPA buffer containing a protease/phosphatase inhibitor cocktail (Cell Signaling Technology). Cell lysates were centrifuged at $14,000 \times g$ for 15 min, and the supernatants were collected and subjected to SDS-PAGE, followed by immunoblotting onto polyvinylidene difluoride membranes. The membranes were blocked with TBS containing 0.1% Tween 20 and 1.0% BSA for 1 h and then probed with primary antibodies. After washing with TBS containing 0.1% Tween 20, the membranes were incubated with horseradish peroxidase-conjugated anti-mouse or anti-rabbit secondary antibodies and visualized with an enhanced chemiluminescence reagent (Perkin Elmer).

### RT-PCR
Total RNA was extracted using an RNA RNeasy Mini Kit (QIAGEN) and reverse-transcribed using the SuperScript IV First-Strand Synthesis System (Thermo Fisher Scientific). cDNAs encoding *SPPL2A* and *SMARCA1* were amplified using specific primers (synthesized by Hokkaido System Science) and Kapa Taq Extra HotStart Ready Mix (Kapa Biosystems). The PCR products were analyzed using the Agilent TapeStation 4150 system with D1000 ScreenTape (Agilent Technologies) for quantification and gel-like image visualization. According to a previous report[95], PSI was calculated as follows: intensity of the exon-included band / (intensity of the exon-included band + intensity of the exon-excluded band). PSI values were calculated using the percentage of integrated area values of respective bands. The sequences of the primers used for RT-PCR are as follows:

*SPPL2A* (forward: ATCATGGTTGAACTCGCAGCT; reverse: TCCAAAACCCAATATTGAAACAGGC), *SMARCA1* (forward: GTCGACTGGATGGACAAACCC; reverse: TAATTCCGAGACCTCCAGCCC).

### Immunoprecipitation from Flp-in T-REx 293 cells
Flp-in T-REx 293 cells expressing full-length WT and K117N *CMTR2* cDNAs and an empty control were established using the Flp-in T-REx 293 system (Thermo Fisher Scientific) according to the manufacturer's protocol. Cells were induced with 0.1 µg/mL doxycycline (Fujifilm) for 24 h to express the target proteins. After induction, immunoprecipitation was performed as described previously[96,97], with some modifications. Briefly, cells were lysed with lysis buffer [20 mM HEPES (pH 7.5), 300 mM NaCl, and 0.5% NP-40] containing a protease inhibitor cocktail and a phosphatase inhibitor cocktail (Roche) and then incubated at 4 °C for 30 min. The insoluble fraction was removed via centrifugation, and the soluble fraction was immunoprecipitated using an anti-FLAG antibody. Precipitated proteins were analyzed by immunoblotting.

### NanoLC-MS/MS analysis
NanoLC-MS/MS was performed based on a previous report[98] with modifications. Samples underwent separation on 12.5% SDS-PAGE gels followed by visualization with a Wako Mass Silver Stain Kit. Gel slices containing proteins were subjected to reduction using 100 mM DTT and subsequent alkylation with 100 mM iodoacetamide. Following washing, tryptic digestion was performed overnight at 30 °C. The resulting peptides were desalted using ZipTip C18 columns (Millipore). Analysis was conducted using nanoLC-MS/MS instrumentation (DiNa HPLC system, KYA TECH Corporation/QSTAR XL Applied Biosystems). Mass spectrometric data were processed using Mascot software.

### Animal experiments
Female BALB/c-nu mice (Charles River, RRID: IMSR_CRL:194; $n = 32$ in total) and C57BL/6 J mice (The Jackson Laboratory, RRID: IMSR_JAX:000664; $n = 80$ in total) aged 5 weeks were used as the xenograft and syngeneic models, respectively. Female mice were used exclusively to minimize aggressive and dominance-related behaviors that could introduce experimental variability. Mice were maintained under standard, strictly controlled specific-pathogen-free conditions: temperature, $22 \pm 0.5$ °C; relative humidity, $55 \pm 10\%$; and a 12-h light/dark cycle (lights on 08:00–20:00). To establish the xenograft model, $1 \times 10^6$ control or $2 \times 10^6$ *CMTR2*-knockout A549 cells were mixed with Matrigel (1:1) and inoculated subcutaneously into BALB/c-nu mice. To establish the syngeneic model, $1 \times 10^6$ LLC cells (control or *CMTR2*-knockout) were inoculated subcutaneously into C57BL/6 J mice. Indisulam was prepared in 15% (2-hydroxypropyl)-β-cyclodextrin and DMSO (Sigma-Aldrich) to yield a stock solution of 2.5 mg/mL. Anti-mouse PD-1 and isotype control antibodies were used for immunotherapy studies. Treatment began 7 days (xenograft) or 3 days (syngeneic) post-inoculation. Mice were randomized into treatment groups. Indisulam (12.5 mg/kg) or vehicle was administered intraperitoneally (IP) 5 days per week. In the syngeneic model, an anti-PD-1 antibody (250 µg) or isotype control antibody was administered IP twice weekly (five doses in total). Tumor volume in the xenograft model was measured three times per week, and that in the syngeneic model was measured twice per week for the first 2 weeks, followed by three times per week thereafter, starting from treatment initiation. Measurements were taken using calipers, and tumor volume was calculated as (width × depth × height × π)/6. Throughout the drug treatment period, mice exhibited no signs of treatment-related toxicity, including body weight loss exceeding 20% or reduced food intake. In vivo measurements were not blinded because blinding was not considered to affect the measurement of results. The Committee for Ethics of Animal Experimentation of the NCC-approved protocol established the maximum allowable tumor diameter for subcutaneous models at 20 mm, which is in accordance with the Japanese Guidelines for Proper Conduct of Animal Experiments (MEXT, MHLW, and MAFF) and the Act on Welfare and Management of Animals. All experimental procedures complied with this ethical guideline, and no tumor exceeded the permitted size threshold.

### Structural modeling and MD simulation of CMTR2 complexed with RNA and SAM
The initial structure of the CMTR2/RNA/SAM complex was prepared as follows. First, after the backbone Cα atoms in the CMTR2 structure predicted by AlphaFold[99] were structurally aligned with those in the CMTR2/mRNA/SAM complex structure modeled by Bujnicki and coworkers[37], named "the Bujnicki model", the Bujnicki model of mRNA/SAM was combined with the AlphaFold structure of CMTR2. Subsequently, the 2′-O position of the first transcribed nucleotide in the mRNA substrate was methylated, given that CMTR1 and CMTR2 sequentially methylate its first and second transcribed nucleotides, respectively. A snRNA substrate was modeled by dimethylating the 2-N position of the N7-methylguanosine (m7G) moiety in

the mRNA substrate. Each of K117N and K275N mutations was introduced into the structure model of WT CMTR2 using the MODELER program[100]. The AMBER ff14SB force field[101] was used for proteins and ions, whereas the OL15 force field[102] was used for RNA. Water was modeled with the TIP3P potential[103]. SAM was protonated to give net charges of 1, reflecting the dominant protonation states at neutral pH. GAMESS was used to optimize the structure of SAM or the m7GpppG moiety in the mRNA/snRNA substrates and to calculate its electrostatic potential at the HF/6-31 G* level[104], after which the atomic partial charges were obtained via the restrained electrostatic potential approach[105]. Other potential parameters of SAM and the m7GpppG moiety in RNA were obtained by the general AMBER force field[106] using the antechamber module of AMBER Tools 12. Approximately 23,000 water molecules were placed around each complex model and approximately 70 sodium and chloride ions (corresponding to 150 mM NaCl) were added to neutralize the system.

According to a previously described procedure[107], MD simulations of the CMTR2/RNA/SAM complex were carried out using the GROMACS 2021.1 program[108]. After energy minimization, each system was equilibrated for 100 ps in a constant number of molecules, volume, and temperature (NVT) ensemble, followed by an MD run of 100 ps in a constant number of molecules, pressure, and temperature (NPT) ensemble with positional restraints applied to the heavy atoms of the protein and RNA. The production runs were conducted under NPT conditions without positional restraints, where the temperature was maintained at 298 K by stochastic velocity rescaling[109] and a Parrinello-Rahman barostat was used to maintain the pressure at 1 bar[110], with the temperature and pressure time constants set to 0.1 and 2 ps, respectively. Five independent production runs of 1 μs (with different atomic velocities) were performed for each *CMTR2* mutant.

### Statistical analysis

Statistical analyses were performed using R ver. 4.2.2 and GraphPad Prism ver. 9.5.1. The two-sided Wilcoxon rank-sum test and two-sided Welch's *t*-test were used to compare numerical variables between unpaired groups. A one-way ANOVA followed by Dunnett's multiple comparison test was used to compare multiple groups against a single control group. A two-way ANOVA was used to analyze the effects of two independent variables (cell type and drug concentration), and the Bonferroni multiple comparisons test was used for post-hoc analysis. A two-way repeated measures ANOVA was used to analyze changes in tumor volume over time. For categorical variables, the two-sided Fisher's exact test was used to compare different groups. PFS data were analyzed by the log-rank (Mantel–Cox) test. Statistical significance for all analyses was set at $P < 0.05$.

### Ethics statement

All research activities described in this study complied with relevant ethical regulations. The study using patient samples from the NCC cohort was approved by the Institutional Review Board of the NCC (IORG0002238) (2005-109). All participants provided written informed consent to participate, and to publish potentially identifiable information (including combinations of indirect identifiers), in accordance with institutional policies. The study protocol for patient-derived organoids was approved by the ethics committee of each facility (approval number: 20110171). All patients in both the NCC cohort and organoid studies provided written informed consent prior to sample collection at all participating institutions. All animal experiments and genetic engineering protocols were approved by the Committee for Ethics of Animal Experimentation of the NCC (approval numbers: A277bM2-23 and A277bM3-25) and the Committee for Ethics of Genetic Modification Experiments of the NCC (approval numbers: B90M3-20 and B90M4-25).

### Reporting summary

Further information on research design is available in the Nature Portfolio Reporting Summary linked to this article.

## Data availability

The RNA-seq data and the whole-exome sequencing data generated in this study have been deposited in the Japanese Genotype–phenotype Archive (JGA) under accession codes JGAS000756 and JGAD000897. Access to the RNA-seq data and the whole-exome sequencing data from the NCC cohort is restricted to ensure patient privacy in accordance with Japanese data privacy laws. Access can be obtained by submitting a research proposal to the National Bioscience Database Center (NBDC), with the accession codes hum0445 [https://humandbs.dbcls.jp/en/hum0445-v1]. Data users shall apply for data use in accordance with the data use application procedures (https://humandbs.biosciencedbc.jp/en/data-use). Data must be used in compliance with NBDC Guidelines for Human Data Sharing, and NBDC Security Guidelines for Human Data (for Data Users), which are available at https://humandbs.dbcls.jp/en/guidelines/data-sharing-guidelines and https://humandbs.dbcls.jp/en/guidelines/security-guidelines-for-users. A summary of how to use controlled access data is available at https://humandbs.dbcls.jp/en/data-use. To request access, contact https://humandbs.ddbj.nig.ac.jp/nbdc/application/. Restrictions on data access and usage are described in the NBDC data sharing guidelines, which are available at https://humandbs.dbcls.jp/en/guidelines/data-sharing-guidelines. The expected timeframe for a response to an access request is within a couple of weeks, and data will be available for the duration of the approved research period. The RNA-seq data from cell line experiments have been deposited in the DDBJ Sequence Read Archive (DRA) under accession number PRJDB18501. The mass spectrometry proteomics data have been deposited in the ProteomeXchange Consortium via the jPOST partner repository under accession code PXD067711. The protein structural data are available in the PDB database under accession code 7B0Y[111] [https://www.rcsb.org/structure/7B0Y], and in the AlphaFold database[92] under accession code Q8IYT2. The chemical structure data used in this study are available in the ChEBI database under accession code CHEBI:167614 [https://www.ebi.ac.uk/chebi/beta/CHEBI:167614]. Publicly available protein–protein interaction data were obtained from STRING v12.0 (https://string-db.org) and BioGRID v4.4 (https://thebiogrid.org)[44,45,112–114]. Source data are provided with the paper. Source data are provided with this paper.

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

## Acknowledgements

We thank M. Shibuya (Division of Pulmonary Medicine, Department of Medicine, Keio University School of Medicine) for technical help with organoid experiments and N. Iida (Division of Genome Analysis Platform Development, NCC Research Institute) for her valuable advice on RNA splicing analysis. We also thank A. Yasui and S. Kanno (Department of Molecular Oncology, IDAC Fellow Laboratory, Institute of Development, Aging and Cancer, Tohoku University) for supervising the proteomic analysis. Proteins interacting with CMTR2 were determined by Japan Proteomics using nanoLC-MS/MS analysis. This work was supported by MEXT via the "Program for Promoting Research on the Supercomputer Fugaku" (Simulation- and AI-driven next-generation medicine and drug discovery based on "Fugaku", JPMXP1020230120) and the FOCUS Establishing Supercomputing Center of Excellence. This research used computational resources of the supercomputer Fugaku provided by the RIKEN Center for Computational Science (Project IDs: hp240211 and hp250220). This work was supported by the Japan Agency for Medical Research and Development (AMED) (JP24ck0106905 to T.K. and JP23ak0101205 and JP24ama221233 to T.N.), the Japan Science and Technology Agency (JST) CREST (JPMJCR1689 to R.H.), and AIP-PRISM (JPMJCR18Y4 to R.H.). This work was also supported by Grants-in-Aid from the Ministry of Education, Culture, Sports, Science, and Technology of Japan (KAKENHI) to S.N. (Grant #22KJ3158), T.N. (Grant #25K02539), and T.K. (Grant #20H00545), from the MSD Life Science Foundation, Public Interest Incorporated Foundation to S.N., by a Research Grant of the Princess Takamatsu Cancer Research Fund to T.N., and by a SGH Cancer Research Grant to T.N.

## Author contributions

S.N., T.N., and T.K. conceptualized and designed the study. S.N., H. Nishinakamura, Y.K., H. Nishikawa, and T.N. designed experimental protocols. S.N., K. Shiraishi, A.M., Y. Shiraishi, H.O., Y.H., Y.Sagae, M.A., and Y.Okuno performed bioinformatics analysis. S.N., E.O., N.T.L., Y. Shimada, K.H., J.H., and A.U. performed wet bench in vitro experiments. E.O., N.T.L., and T.N. performed in vivo mouse experiments. K. Shiraishi, K.O., K. Sugihara, H.Y., Y.Ohe, S.W., Y.Y., R.H., and T.K. performed patient sample acquisition. S.E., J.M., T.Y., and Y.G. collected patient clinical information. The original manuscript draft was written by S.N., with review and editing by K. Shiraishi, A.M., T.N., T.K., and M.A.

## Competing interests

Y.Y., Y.G., and T.K. have financial relationships with Thermo Fisher Scientific, whose products were used in this study. Y.Y. has received research grants, Y.G. has received payment for lectures, and T.K. has received patent fees from this company. The other authors declare no competing interests.

## Additional information

Shigenari Nukaga [1,2], Kouya Shiraishi [1,3], Kenta Hamabe[2], Akifumi Mochizuki [1], Yu Hamaguchi [1], Emi Ogawa[1], Nguyen Thai Le [1], Yoko Shimada[1], Hanako Ono [3], Hitomi Nishinakamura [4], Yoshihisa Kobayashi [5], Junko Hamamoto [2], Ayako Ui[6], Mitsugu Araki [7], Yukari Sagae [7], Keiko Ohgino [2], Kai Sugihara [8], Satoshi Endo [1], Jun Miyakoshi [1], Yuichi Shiraishi [9], Hiroyuki Yasuda[2], Yasushi Okuno [7], Tatsuya Yoshida [10], Yasushi Goto[10], Yuichiro Ohe[10], Shun-Ichi Watanabe[11], Yasushi Yatabe [5,12], Hiroyoshi Nishikawa [4], Ryuji Hamamoto [13], Takashi Kohno [1] ✉ & Takashi Nakaoku [1] ✉

[1]Division of Genome Biology, National Cancer Center Research Institute, Chuo-ku, Tokyo, Japan. [2]Division of Pulmonary Medicine, Department of Medicine, Keio University School of Medicine, Shinjuku-ku, Tokyo, Japan. [3]Department of Clinical Genomics, National Cancer Center Research Institute, Chuo-ku, Tokyo, Japan. [4]Division of Cancer Immunology, National Cancer Center Research Institute, Chuo-ku, Tokyo, Japan. [5]Division of Molecular Pathology, National Cancer Center Research Institute, Chuo-ku, Tokyo, Japan. [6]Department of Molecular Oncology, IDAC Fellow Laboratory, Institute of Development, Aging and Cancer, Tohoku University, Sendai, Miyagi, Japan. [7]Graduate School of Medicine, Kyoto University, Kyoto, Japan. [8]Division of Pulmonary Medicine, Kawasaki Municipal Hospital, Kawasaki-ku, Kanagawa, Japan. [9]Division of Genome Analysis Platform Development, National Cancer Center Research Institute, Chuo-ku, Tokyo, Japan. [10]Department of Thoracic Oncology, National Cancer Center Hospital, Chuo-ku, Tokyo, Japan. [11]Department of Thoracic Surgery, National Cancer Center Hospital, Chuo-ku, Tokyo, Japan. [12]Department of Diagnostic Pathology, National Cancer Center Hospital, Chuo-ku, Tokyo, Japan. [13]Division of Medical AI Research and Development, National Cancer Center Research Institute, Chuo-ku, Tokyo, Japan. ✉e-mail: tkkohno@ncc.go.jp; tnakaoku@ncc.go.jp

