## [Transparent Peer Review file · Nature Communications]

Mutation of *CMTR2* in Lung Adenocarcinoma Alters RNA Alternative Splicing and Reveals Therapeutic Vulnerabilities

Corresponding Author: Dr Takashi Nakaoku

Version 0:

Reviewer comments:

Reviewer #1

(Remarks to the Author)

The study entitled “Mutation of *CMTR2* in Lung Adenocarcinoma Alters RNA Alternative Splicing and Reveals Therapeutic Vulnerabilities” by Nukaga et al. analyses mutations and alternative splicing in 751 lung adenocarcinoma samples from Japanese National Cancer Center cohort, combined with 519 samples from The Cancer Genome Atlas. Nukaga and colleagues used the t-distributed stochastic neighbor embedding (t-SNE) statistical method to visualize splicing patterns and observed significant inter-tumor heterogeneity. A key finding of the study is the identification of a molecular subtype of lung adenocarcinoma that is specifically associated with inactivating mutations in the *CMTR2* gene. *CMTR2* encodes an enzyme involved in the methylation of mRNA and small nuclear RNA (snRNA). As been previously shown by Campbell et al, Nature Genetics 2016, the study confirms that mutations in *CMTR2* gene occur in approximately 4-6% of lung adenocarcinoma samples, and these mutations are predominantly truncating mutations. The study further demonstrates that these mutations lead to alternative splicing including skipped exons in specific genes, and that they sensitize cancer cells to treatments to sulfonamide-based RNA splicing modulators. The authors also claim that *CMTR2* mutation increases sensitivity to immune checkpoint blockade therapy although the supporting evidence here is quite weak. Overall, this is a very interesting paper that demonstrates endogenous functions of *CMTR2*. The manuscript fits the scope of Nature Communications.

Our specific questions and suggestions are as follow:

On page 3 line 70 authors introduce their NCC cohort as National Cancer Center Hospital (NCCH). However, throughout the text authors refer to it as NCC. For clarity and consistency, we recommend the following change: “National Cancer Center (NCC) Hospital”.

What are the expression levels of *CMTR2* in the *CMTR2* mutated cluster (Fig. 1h) compared to others in the NCC ADC cohort? Supplementary Fig. 3a shows the most frequently mutated genes in the NCC ADC cohort, but it also would be interesting to see analysis of DEGs as well as enrichment analysis in *CMTR2* cluster, which could further shed light on the role of *CMTR2* in splicing.

Figures 1i, 1j, and 1k are probably un-necessary. Figure 1i is highly complex and not informative as a main text figure. The results of Figure 1j have been previously described (Smietanski et al., Nature Communications, 2014). Similarly, the key results of Figure 1k (frequent inactivating mutation of *CMTR2*) have been previously described (Campbell et al., Nature Genetics, 2016). Thus the sentence “In summary, these finding provide novel insight into the role of *CMTR2* mutations in LADC and establish *CMTR2* as a putative tumor suppressor gene targeted by smoking-associated mutations” would be more accurately described as: “In summary, these findings provide novel insight into the role of *CMTR2* mutations in LADC and confirm previous reports of *CMTR2* as a putative tumor suppressor gene, adding the observation that *CMTR2* mutations are mostly smoking-associated”.

Supplementary Fig. 3a uses the same colors (red and blue) for mutation type, smoking status, and driver genes making it difficult to interpret the panel. We suggest using different colors for the subgroups.

Supplementary Fig. 3c: the plot of *CMTR2*-mutated samples obtained from TCGA cancer types shows that high *CMTR2* mutation frequency is associated with hypermutated subtypes characterized by *POLE* mutations or microsatellite instability-high (MSI-H). However, this this association was not observed in lung cancer. The *CMTR2* missense mutations observed in

high TMB settings are likely enriched for passenger mutations (line 143 to 152 section), perhaps in contrast to the lung cancer missense mutations described in Figure 2 below. This should be commented by the authors.

Page 5, line 152 figure/data reference is missing for this statement about independent cohorts.

Figure 2. To further support the conclusion of the importance of the CMTR2 mutations in the non-catalytic domain in the methylation activity of CMTR2, we recommend referencing the study by Smietanski et al. Nat Commun 2014, in particular the mutagenesis analysis which shows that both domains of CMTR2 are required for enzymatic activity of the CMTR2 protein.

Figure 3. suggests that CMTR2 mutations induce splicing changes similar to the splicing factor genes U2AF1 and RBM10. Do CMTR2 mutations co-occur with U2AF1 or RBM10 mutations in NCC and TCGA ADC cohort? (is it addressed in Supplementary Table 1 and 2?)

Supplementary Fig. 6 Western blot confirming KO of Cmtr2 in LCC cell line is missing (assuming the Cmtr2 antibody against mouse Cmtr2 is good).

Figure 4. A little more explanation is needed why authors focus specifically on SPPL2A and SMARCA1 events (current text says they are "representative SE event". Are they most highly enriched SE events? The list of all 71 SE events common to both human tumor samples and KO cell lines would be helpful.

Figure 4a is Venn diagram showing overlap of significantly differential SE events in CMTR2 mutant vs wild-type samples in TCGA and NCC cohorts and CMTR2 KO cell lines. We suggest including also DEGs (and gene ontology and KEGG pathway enrichment analysis of the DEGs) of CMTR2-mutated samples in NCC and TCGA cohorts and cell lines.

Figure 4 e-h, Supplementary Fig. 8b Images lack quantification/ statistics

Figure 4 e-g, Supplementary Fig. 8a. We do acknowledge that measuring Cap2 methylation status is technically challenging, but, in ideal world, it would be great to confirm CMTR2 is acting as an RNA cap methyltransferase in the cell lines.

Fig. 4i shows the increase exon inclusion in SPPL2A was observed in multiple cancer types in TCGA dataset consistent with findings in Fig 4a-h. Does SMARCA1 has decrease exon inclusion across different tumor types in TCGA? Is this plot needed?

Figure 4K is a model but it does not support the protein-protein interaction analysis described in the text. The analysis in Supplementary Figure 11a and 11b is also not evaluable. We recommend removing these figures. The data in Supplementary Tables 5 and 6 appear reasonable and support the conclusions described; that should be sufficient.

Line 245, "Next, we investigated whether CMTR2 deficiency is a potential therapeutic target in lung cancer". I don't think one can call deficiency a therapeutic target. This might be better phrased as something like "Next, we investigated whether CMTR2 deficiency is associated with potential therapeutic vulnerabilities in lung cancer".

The analysis of indisulam sensitivity in CMTR2 knockout is very interesting (Figure 5 and Supplementary Figure 12).

Fig. 5h would benefit from quantification/ statistics

Figure 5m/5n should have a Western blot for CMTR2. Maybe these are the same cells shown in Supplementary Figure 8a? If so, the authors need to say so. If not, they need to do additional Western blots.

Fig. 6 is probably the weakest part of the paper at this point. The authors show a modest cooperative effect of the combination of indisulam and anti-PD1 antibodies and also some very modest degrees of responses to immune checkpoint blockade in CMTR2 mutant tumors. These data are intriguing and deserving of further development but probably not suitable for publication at this time.

In Discussion on page 9, lines 321-323. the authors state that involvement of CMTR2 in carcinogenesis was suggested in previous studies referencing several papers. The authors should include the reference to the study by Campbell et al, Nature Genetics 2016 that was the first one to identify CMTR2 / FTSJD1 as a recurrently mutated tumor suppressor gene in lung ADC in both introduction and discussion.

The authors should also discuss recent publications on CMTR2 including the recent evidence for adenosine modifications by CMTR2 as an alternative/addition to guanosine methylation (<https://pubmed.ncbi.nlm.nih.gov/39993262/>), the identification of CMTR2 polymorphisms as germline risk alleles for lung cancer (<https://pubmed.ncbi.nlm.nih.gov/39472694/>), and the evidence for a role of Cmtr2 in embryonic development (<https://pubmed.ncbi.nlm.nih.gov/39094818/>) (<https://pubmed.ncbi.nlm.nih.gov/37436893/>)

Throughout: The "NCI-H" cell lines are described as "H" cell lines. This should be corrected. For example, H3122 should be NCI-H3122, H1373 should be NCI-H1373 and so forth throughout the text.

Reviewer #2

(Remarks to the Author)

Reviewer #3

(Remarks to the Author)

Review of "Mutation of CMTR2 in Lung Adenocarcinoma Alters RNA Alternative Splicing and Reveals Therapeutic Vulnerabilities"

Summary and Key Findings

This study provides an in-depth analysis of alternative splicing in lung adenocarcinoma (ADC), highlighting CMTR2 mutations as a potential driver of splicing disruption. The authors demonstrate that:

Alternative splicing (AS) can serve as a feature to understand tumour biology, capturing histological subtypes and reflecting underlying genomic alterations.

Specific splicing factors, including CMTR2, are recurrently mutated in lung ADC, with these mutations influencing distinct splicing patterns in large independent cohorts ($n > 1000$).

CMTR2, a key factor in 5' mRNA capping, is frequently mutated in smokers. Truncating mutations may function as tumour suppressor gene (TSG) drivers, given their association with global splicing alterations.

CMTR2 loss results in widespread splicing dysregulation, which is partially reversible with CMTR2 overexpression.

CMTR2 mutations extend beyond lung cancer, predominantly appearing in endometrial cancer and melanoma.

CMTR2-mutant cancers exhibit therapeutic vulnerabilities, including sensitivity to sulfonamide therapy (indisulam) and immune checkpoint blockade (ICB), potentially due to the generation of immunogenic neoantigens.

Significance and Context

The findings hold significant implications for cancer biology and therapeutics. While CMTR2 mutations are relatively infrequent ($\leq 6\%$ in lung, breast, and endometrial cancers, and lower in others), the overarching concept of targeting splicing disruption as a cancer vulnerability is highly relevant. This aligns with emerging literature on splicing-targeted therapies, although further validation is required to establish its translational potential.

Areas Requiring Further Investigation

Mechanistic Link Between CMTR2 and Splicing Dysregulation

The precise role of CMTR2 in splicing remains unclear. Given that CMTR2 primarily functions in mRNA capping, it is not immediately evident how its loss directly perturbs splicing. The authors should clarify whether the observed splicing defects arise from direct CMTR2 involvement or if secondary mechanisms (e.g., destabilisation of specific splicing regulators) are at play.

Further functional assays could establish whether CMTR2 interacts with splicing machinery or indirectly affects splicing through changes in transcript stability.

Validation of Indisulam Sensitivity and Synthetic Lethality with RBM39

The synthetic lethal relationship with indisulam is inferred based on RBM39 degradation. However, indisulam is known to have off-target effects. It would be valuable to demonstrate that RBM39 knockout (KO) alone recapitulates the CMTR2 loss phenotype, thereby confirming the specificity of this interaction.

Data Analysis and Interpretation

Short-Read Sequencing Limitations: The study relies on rMATS for AS event detection, which is constrained by the inherent limitations of short-read RNA sequencing. Long-read sequencing should be employed for validation, or at minimum, the authors should acknowledge the potential bias and incomplete detection of splicing events, especially those involving complex translocations.

tSNE Representation and Cluster Annotations:

While the PSI-based tSNE captures histological features, the naming of clusters in Figure 1D may be misleading. The figure suggests a strict one-to-one relationship between clusters and features (e.g., splicing factor mutations, epithelial-mesenchymal transition [EMT] phenotype, neuroendocrine differentiation). However, feature expression within tSNE clusters is not comprehensively visualised.

Specific mutations (e.g., U2AF1, RBM10) appear in multiple locations, but only a subset is highlighted in Figure 1D. A more transparent depiction of feature distribution across the tSNE space would improve interpretability.

Correction of Specific Claims:

Line 97: "Other major histological subtypes displayed distinct distribution" should be revised, as minor subtypes are dispersed within the ADC cluster.

Line 115: "Gastric lineage" should be replaced with "gastrointestinal lineage."

Methodology and Reproducibility

The methodology is largely sound but should incorporate additional validation steps:

The accuracy of AS detection should be assessed using long-read sequencing, as this is now the standard for AS analysis.

The selection criteria for highly variable PSI events in PCA/tSNE should be explicitly stated to ensure unbiased dimensionality reduction.

Overall, the methods section provides sufficient detail for reproducibility.

Figure-Specific Comments

Figure 1

Panel A (Workflow using rMATS)

The reliance on short-read sequencing should be acknowledged as a limitation.

While rMATS is unlikely to over-call splice variants, it may under-detect certain splicing events, particularly those involving trans-splicing or complex rearrangements.

Panel C (tSNE Clustering of LADC)

Clarify how the PSI matrix was reduced to tSNE.

Ensure that dimensionality reduction was performed in an unbiased manner (i.e., which PSI values were included in PCA?).

Panel D (PSI tSNE with Feature Enrichment)

Demonstrate the actual distribution of splicing factor mutations, EMT markers, and neuroendocrine markers across the tSNE rather than relying solely on categorical cluster labels.

Improve cluster labelling to reflect the heterogeneous nature of the data.

Supplementary Figure 1

Panel A (Clustering by PSI)

Clustering appears robust, but the claim that "other major histological subtypes displayed distinct distribution" is questionable.

Panel G (GSEA of GI-related pathways)

Revise "gastric lineage" to "gastrointestinal lineage."

Supplementary Figure 2

Panels A-D (EMT Phenotype Validation)

Boxplots alone do not fully demonstrate the spatial distribution of EMT features. Overlaying EMT markers on the tSNE plot would provide stronger support for this claim.

Panel D (GSEA for EMT Cluster)

Further validation of EMT assignment within tSNE is warranted.

Panels E-F (Neuroendocrine Gene Expression)

Similar to EMT analysis, consider mapping neuroendocrine markers onto the tSNE plot.

Reviewer #4

(Remarks to the Author)

In this manuscript Nakaoku and colleagues investigate the RNA splicing landscape in lung adenocarcinoma. Using large RNAseq datasets they identify different splicing subtypes associated with mutations in a number of different RNA splicing regulators. This includes the novel identification of splicing modifications in tumours carrying mutated CMTR2. They further go on to confirm the role of CMTR2 in mediating RNA splicing using a variety of model systems. Further, they show that CMTR2 mutation sensitises tumours to broader RNA splicing inhibition. Finally, they demonstrate that CMTR2 mutation sensitises to immunotherapy in mouse models and potentially marks patients responsive to immunotherapy. Overall, the findings are novel, interesting and likely impactful. In particular, the sensitisation of CMTR2 mutant tumours to both anti-splicing therapy and/or immunotherapy could lead to novel treatments and better patient stratification. I have a number of minor points the authors should address prior to publication.

- 1) The authors identify CMTR2 as a putative tumour suppressor but there is no direct analysis of the effects of CMTR2 deletion on tumour cell growth. Can the authors investigate whether CMTR2 deletion increases tumourigenic phenotypes in vitro and in vivo?
- 2) This is particularly relevant given the immunogenic phenotype following CMTR2 deletion, which would be expected to suppress tumour initiation and growth. What is the proposed mechanism of action of CMTR2 deletion?
- 3) In Figure 4, the authors investigate splicing changes induced by loss of CMTR2. Are any functional processes enriched in these altered splicing events? And could these indicate potential mechanism of CMTR2 deletion driving tumourigenesis?
- 4) Can the phenotypic effects in the PDO model (Figure 5), be rescued by reintroducing wild-type CMTR2?
- 5) In Fig 6F the authors conclude that CMTR2 mutation is predictive of a good response to immune checkpoint blockade but only the CMTR2 mutated samples are shown. What is the responsiveness in CMTR2 non-mutated tumours so an accurate comparison of response rate can be carried out?

Reviewer #6

(Remarks to the Author)

Version 1:

Reviewer comments:

Reviewer #1

(Remarks to the Author)

The authors have satisfactorily addressed our comments and suggestions. We have no further comments for the revised manuscript. We believe it presents a compelling observation, investigating the RNA splicing landscape in lung adenocarcinoma and highlighting CMTR2 mutations as a potential driver of splicing disruption.

Reviewer #2

(Remarks to the Author)

Reviewer #3

(Remarks to the Author)

We would like to acknowledge the considerable work that has gone into addressing the earlier concerns. Overall, the revisions are thorough and have significantly strengthened the manuscript.

The concern about the lack of mechanistic insight into how truncating mutations in CMTR2 lead to splicing alterations has been addressed particularly well. The new data showing that CMTR2 physically interacts with components of the splicing machinery, and that this interaction is lost when the gene is truncated, provide a much clearer picture of its role. The accompanying mRNA-level confirmation of splicing effects further supports the proposed mechanism and adds weight to the biological relevance of the findings.

Regarding validation of the splicing changes using long-read sequencing: although the ONT direct RNA sequencing results have not been incorporated into a main figure, it is good to see that the authors have performed this work. The relevant details (lines 253–256 and SI Fig. 10) suggest the long-read data generally support the conclusions drawn from short-read analyses. The BAMBU analysis of target genes indicating differential isoform expression following CMTR2 knockout is in line with expectations. However, a direct comparison between short- and long-read data—ideally side-by-side—would have been helpful in fully evaluating concordance between these approaches. The shortcomings of not using long-read sequencing throughout the study should be discussed in terms of false positives and limitations to this work within the discussion

Reviewer #4

(Remarks to the Author)

The authors have now addressed my concerns.

Reviewer #6

(Remarks to the Author)

[Point-by-point responses to the reviewers]

“Mutation of CMTR2 in Lung Adenocarcinoma Alters RNA Alternative Splicing and Reveals Therapeutic Vulnerabilities”

Authors: Nukaga et al.

We thank the editor and reviewers for their thoughtful and thorough critiques, which have greatly improved our manuscript. Below, we provide point-by-point responses to their comments. All changes made to the manuscript are marked with additions highlighted in yellow. Response only data figures are provided in this letter to support our responses and are not part of the revised manuscript.

Reviewer #1

Reviewer Comments:

1. *On page 3 line 70 authors introduce their NCC cohort as National Cancer Center Hospital (NCCH). However, throughout the text authors refer to it as NCC. For clarity and consistency, we recommend the following change: “National Cancer Center (NCC) Hospital”.*

Response: Thank you for this important suggestion. We have revised the manuscript on page 3 lines 75–76 to consistently refer to our cohort as "National Cancer Center (NCC) Hospital" throughout the text.

2. *What are the expression levels of CMTR2 in the CMTR2 mutated cluster (Fig. 1h) compared to others in the NCC ADC cohort? Supplementary Fig. 3a shows the most frequently mutated genes in the NCC ADC cohort, but it also would be interesting to see analysis of DEGs as well as enrichment analysis in CMTR2 cluster, which could further shed light on the role of CMTR2 in splicing.*

Response: You have asked an important question. We compared CMTR2 expression levels between the CMTR2-mutated cluster and other samples in the NCC cohort, but did not observe a significant decrease in CMTR2 expression in the former group (Response only data Fig. 1a). By contrast, CMTR2 expression levels were decreased in the CMTR2-mutated cluster in the TCGA dataset (Response only data Fig. 1b). The estimated tumor content based on our NCC whole-exome sequencing data was lower than the published estimated tumor content in the TCGA, which may explain why we were unable to detect a significant difference in our cohort (Response only data Fig. 1c, d).

We appreciate your suggestion regarding DEG analysis in the CMTR2 cluster. Following your recommendation, we have performed differential gene expression analysis and pathway enrichment analysis using GO (Molecular Function, Biological Process, and Cellular Component) and KEGG gene sets comparing CMTR2 cluster and non-cluster samples in the NCC cohort. However, we did not identify statistically significant enrichment of gene sets

that could suggest functional implications. This may be due to the dilution effect of non-tumor components in clinical specimens, which potentially mask the impact of CMTR2 on gene expression. Additionally, following the same approach as was used to extract common SE events in Fig. 4a, we identified DEGs common to the NCC and TCGA cohorts as well as to A549 and NCI-H3122 CMTR2-knockout cell lines (Response only data Fig. 2a, b). We performed pathway enrichment analysis of these genes, but no significant pathway enrichment was observed. This result may also be attributable to tumor purity issues in clinical specimens. In Fig. 5g, we demonstrated significant changes in expression of cellular components involved in RNA splicing in A549 and LLC CMTR2-knockout cells. However, we did not observe significant enrichment of gene sets related to molecular functions (GOMF) or biological processes (GOBP). These findings suggest it is challenging to infer CMTR2 inactivation function from gene expression data, and further investigation is needed. We have included this information in the Discussion section (page 11, lines 383–388).

Response only data Fig. 2

3. Figures 1i, 1j, and 1k are probably un-necessary. Figure 1i is highly complex and not informative as a main text figure. The results of Figure 1j have been previously described (Smiotanski et al., Nature Communications, 2014). Similarly, the key results of Figure 1k (frequent inactivating mutation of CMTR2) have been previously described (Campbell et al., Nature Genetics, 2016). Thus the sentence "In summary, these findings provide novel insight into the role of CMTR2 mutations in LADC and establish CMTR2 as a putative tumor suppressor gene targeted by smoking-associated mutations" would be more accurately described as: "In summary, these findings provide novel insight into the role of CMTR2 mutations in LADC and confirm previous reports of CMTR2 as a putative tumor suppressor gene, adding the observation that CMTR2 mutations are mostly smoking-associated".

Response: Thank you for your constructive comments. We agree with your assessment of Fig. 1i and have removed it due to its limited value. Regarding Fig. 1j, while the results have been previously described by Smietanski et al. (*Nature Communications* 2014), we believe its inclusion is valuable to enhance readers' understanding because CMTR2 is not widely recognized among lung cancer researchers. Therefore, we have retained this figure as Fig. 1h.

With respect to Fig. 1k, we acknowledge the seminal work by Campbell et al. (*Nature Genetics* 2016), which first reported that LADC exhibits significantly more CMTR2 mutations than pan-cancer, with an enrichment of inactivating mutations. Therefore, we referred to this study as reference 11 in the text (page 3, lines 84–85 and page 11, lines 368–369). While their study did not detail the specific mutation counts or types for individual cancer types, our study demonstrates that while some cancers with frequent CMTR2 mutations have hypermutated status, LADC does not, and CMTR2 mutations in LADC are associated with smoking background, resulting in a high frequency of inactivating mutations. We believe it is important to present the distribution of CMTR2 mutation counts and types across the TCGA pan-cancer cohort to understand this. Therefore, we have retained this figure as Fig. 1i.

We have revised the summary sentence as you suggested to more accurately reflect the nature of our findings (lines 181–183).

4. *Supplementary Fig. 3a uses the same colors (red and blue) for mutation type, smoking status, and driver genes making it difficult to interpret the panel. We suggest using different colors for the subgroups.*

Response: Thank you for your valuable comment. As suggested, we have revised Supplementary Fig. 3a using different colors for the subgroups to improve its clarity and interpretability.

5. *Supplementary Fig. 3c: the plot of CMTR2-mutated samples obtained from TCGA cancer types shows that high CMTR2 mutation frequency is associated with hypermutated subtypes characterized by POLE mutations or microsatellite instability-high (MSI-H). However, this association was not observed in lung cancer. The CMTR2 missense mutations observed in high TMB settings are likely enriched for passenger mutations (line 143 to 152 section), perhaps in contrast to the lung cancer missense mutations described in Figure 2 below. This should be commented by the authors.*

Response: Thank you for bringing this important point to our attention. We have added the following sentence (page 5, lines 166–167): " These results suggest that while CMTR2 mutations may be passenger mutations in hypermutated subtype cancers, they appear to have functional significance in lung cancer."

6. Page 5, line 152 figure/data reference is missing for this statement about independent cohorts.

Response: Thank you for pointing out this missing reference. We apologize for the insufficient data presentation. We have added the analysis results from the IntOGen web platform as Supplementary Table 3 to support our statement about independent cohorts.

7. Figure 2. To further support the conclusion of the importance of the CMTR2 mutations in the non-catalytic domain in the methylation activity of CMTR2, we recommend referencing the study by Smietanski et al. Nat Commun 2014, in particular the mutagenesis analysis which shows that both domains of CMTR2 are required for enzymatic activity of the CMTR2 protein.

Response: Thank you for highlighting this important point. Upon reviewing the study by Smietanski et al. (*Nature Communications* 2014), we noted that while their mutagenesis experiments were conducted only on the catalytic domain, they demonstrated that truncated CMTR2 proteins lacking the non-catalytic domain have impaired methylation function. This supports the importance of the non-catalytic domain, which we have now addressed (page 6, lines 206–210).

8. Figure 3. suggests that CMTR2 mutations induce splicing changes similar to the splicing factor genes U2AF1 and RBM10. Do CMTR2 mutations co-occur with U2AF1 or RBM10 mutations in NCC and TCGA ADC cohort? (is it addressed in Supplementary Table 1 and 2?)

Response: We appreciate your insightful question about the co-occurrence of mutations. We have added an oncoplot showing the co-occurrence of CMTR2, RBM10, and U2AF1 mutations in both the NCC and TCGA cohorts as Supplementary Fig. 7a, b. Additionally, we have included information about splicing factor mutations that co-occur with CMTR2 mutations in the column titled "Compound Spliceosomal Mutation" of Supplementary Tables 1 and 2. These results demonstrate that CMTR2 mutations rarely co-occur with RBM10 and U2AF1 mutations, indicating that splicing abnormalities in CMTR2-mutated samples are not influenced by these other splicing factors and that CMTR2 mutations independently induce splicing abnormalities similar to mutations of other splicing factors. We have added the following text (page 7, lines 221–223): "Additionally, similar to the mutual exclusivity of major splicing factor mutations in solid and hematological malignancies, CMTR2 mutations rarely co-occurred with RBM10 and U2AF1 mutations in both cohorts (Supplementary Fig. 7a, b and Supplementary Tables 1, 2)."

9. Supplementary Fig. 6 Western blot confirming KO of Cmtr2 in LCC cell line is missing (assuming the Cmtr2 antibody against mouse Cmtr2 is good).

Response: Thank you for your valuable comment regarding the Western blot confirmation of Cmtr2 knockout in the LCC cell line. The anti-CMTR2 antibody (Thermo Fisher Scientific, PA5-61696) used in our study was generated against a human-derived epitope that shares 76% sequence homology with its mouse ortholog according to the product information. Due

to this difference in homology, while we were able to clearly detect CMTR2 protein in human lung cancer cell lines, we could not obtain sufficient signal in mouse cell lines (Response only data Fig. 3). Despite testing multiple antibodies, we were unable to detect specific bands for CMTR2 protein in mouse cells.

Response only data Fig. 3

10. *Figure 4. A little more explanation is needed why authors focus specifically on SPPL2A and SMARCA1 events (current text says they are “representative SE event”. Are they most highly enriched SE events? The list of all 71 SE events common to both human tumor samples and KO cell lines would be helpful.*

Response: Thank you for your insightful comment. We have added the complete list of all 71 shared SE events as Supplementary Data 1. Furthermore, we have included a volcano plot in Supplementary Fig. 9a that illustrates the differentially spliced SE events between *CMTR2* mutant and WT samples in the NCC cohort that were also consistently observed across other cohorts and knockout cell lines.

Our selection of *SPPL2A* and *SMARCA1* for representative SE events was based on several criteria. Specifically, they exhibit relatively large changes in PSI values (delta PSI) and contain sufficiently long skipped exons that can be clearly visualized by PCR validation experiments. We have added the following explanation (page 7, lines 247–251): " Among these shared splicing events, *SPPL2A* (Signal Peptide Peptidase-Like 2A) and *SMARCA1* (SWI/SNF-Related,

Matrix-Associated, Actin-Dependent Regulator of Chromatin, Subfamily A, Member 1) were selected for further validation due to their robust PCR-detectable exon length; SPPL2A showed increased exon inclusion and SMARCA1 showed increased exon exclusion (Fig. 4b and Supplementary Fig. 9b–e)."

11. *Figure 4a is Venn diagram showing overlap of significantly differential SE events in CMTR2 mutant vs wild-type samples in TCGA and NCC cohorts and CMTR2 KO cell lines. We suggest including also DEGs (and gene ontology and KEGG pathway enrichment analysis of the DEGs) of CMTR2-mutated samples in NCC and TCGA cohorts and cell lines.*

Response: Thank you for this insightful suggestion. We have already addressed this comprehensive analysis in our response to Comment #2. Please refer to that response for our detailed findings.

12. *Figure 4 e-h, Supplementary Fig. 8b Images lack quantification/ statistics*

Response: Thank you for this important comment. To address the lack of quantification in the original submission, we have made several improvements to our analysis and presentation as detailed below.

We have replaced the conventional gel electrophoresis images of PCR products with analysis using the Agilent TapeStation system, which has superior quantitative capabilities. The figures now display gel-like images generated by the TapeStation system. We have also added quantitative PSI values calculated from band signal intensities.

PSI values were calculated according to the established method described in reference 95 using the following formula: $PSI = \text{intensity of the exon-included band} / (\text{intensity of the exon-included band} + \text{intensity of the exon-excluded band})$.

For both the knockout experiments shown in Fig. 4e and Supplementary Fig. 11 and the CMTR2 overexpression experiments shown in Supplementary Fig. 12, we have performed statistical comparisons of PSI changes between different conditions using data from multiple independent experiments (n=3). The statistical analysis and quantitative results are included in the revised figures and their corresponding legends.

13. *Figure 4 e-g, Supplementary Fig. 8a. We do acknowledge that measuring Cap2 methylation status is technically challenging, but, in ideal world, it would be great to confirm CMTR2 is acting as an RNA cap methyltransferase in the cell lines.*

Response: We appreciate your valuable feedback. Due to technical limitations, we were unable to experimentally demonstrate that CMTR2 functions as an RNA cap methyltransferase in the cell lines used in this study. However, a previous study by Despic et al. (*Nature* 2023, ref. 27) that used CLAM-Cap-seq to evaluate RNA methylation showed that CMTR2 knockout in HEK293T cells results in almost complete loss of Cap2 methylation. These

data demonstrate that CMTR2 is essential for Cap2 formation in human cells, suggesting that Cap2 methylation is also deficient in the knockout cell lines used in our study.

We have explicitly stated this limitation in the Discussion section (page 12, lines 427–429).

14. *Fig. 4i shows the increase exon inclusion in SPPL2A was observed in multiple cancer types in TCGA dataset consistent with findings in Fig 4a-h. Does SMARCA1 has decrease exon inclusion across different tumor types in TCGA? Is this plot needed?*

Response: Thank you for your insightful question regarding *SMARCA1* splicing patterns across tumor types. In response to your query, we have added Fig. 4j, which shows the PSI values of *SMARCA1* SE events across different tumor types in the TCGA dataset, and Supplementary Fig. 13b, which illustrates *SMARCA1* PSI values in normal tissues and cell lines. Our analysis confirmed that exon inclusion of *SMARCA1* was decreased in *CMTR2* mutant samples across various cancer types. We appreciate your suggestion because it prompted us to include additional supporting data for our findings. We have incorporated this information in the main text (page 8, lines 261–264).

15. *Figure 4K is a model but it does not support the protein-protein interaction analysis described in the text. The analysis in Supplementary Figure 11a and 11b is also not evaluable. We recommend removing these figures. The data in Supplementary Tables 5 and 6 appear reasonable and support the conclusions described; that should be sufficient.*

Response: Thank you for this thoughtful comment. We have substantially addressed these concerns by adding comprehensive experimental data to support the mechanistic model depicted in Fig. 5f (Fig. 4k in the original submission).

We have performed immunoprecipitation-mass spectrometry experiments using Flag-tagged CMTR2 constructs in Flp-in T-REx 293 cells (Fig. 5a–c). These experiments directly demonstrated that WT CMTR2 physically interacted with U1 snRNP proteins, while the CMTR2 K117N mutant, which is considered a loss-of-function variant, exhibited a significantly reduced binding capacity (Fig. 5d, e and Supplementary Data 2). This experimental evidence validates the protein–protein interaction predicted by our database analysis (Supplementary Tables 6 and 7). Consistently, the 71 shared differentially spliced SE events predominantly exhibited GT-AG splice sites typical of major introns targeted by U1 snRNP (Supplementary Fig. 14). This finding reinforces the functional relevance of CMTR2-U1 snRNP interactions in splicing regulation.

These results suggest that CMTR2 deficiency potentially affects splicing through mechanisms related to interactions between CMTR2 and snRNPs. We have incorporated these revisions into the Results (pages 8–9, lines 268–291) and Discussion (page 11, line 395) sections. Additionally, we have expanded the Methods section to include detailed protocols for the newly added experiments in the sub-sections titled “Immunoprecipitation from Flp-in T-REx

293 cells" (page 19, lines 674–682) and "NanoLC-MS/MS analysis" (pages 19–20, lines 684–691).

16. *Line 245, "Next, we investigated whether CMTR2 deficiency is a potential therapeutic target in lung cancer". I don't think one can call deficiency a therapeutic target. This might be better phrased as something like "Next, we investigated whether CMTR2 deficiency is associated with potential therapeutic vulnerabilities in lung cancer".*

Response: We appreciate your insightful comment regarding our phrasing. In accordance with your suggestion, we have revised the text (page 9, lines 294 to 295) to read "Next, we investigated whether CMTR2 deficiency is associated with potential therapeutic vulnerabilities in lung cancer".

17. *The analysis of indisulam sensitivity in CMTR2 knockout is very interesting (Figure 5 and Supplementary Figure 12).*

Response: Thank you for your positive feedback on our analysis of indisulam sensitivity in CMTR2-knockout cells. These findings highlight the potential therapeutic implications of CMTR2 deficiency and provide a foundation for future investigations into targeted approaches for lung cancer treatment.

18. *Fig. 5h would benefit from quantification/ statistics*

Response: Thank you for this valuable suggestion. In response to this comment, we have performed additional experiments with proper quantification and statistical analysis. We have conducted colony formation assays using A549 control cells and three independent A549 CMTR2-knockout clones (#1, #2, and #3) with n=4 replicates for each cell line at each indisulam concentration. Colony areas were measured and subjected to statistical analysis using appropriate comparative tests. Colony formation capacity was significantly suppressed in all CMTR2-knockout clones compared with control cells upon indisulam treatment. We have updated Fig. 6h with representative images from these quantified experiments and added Fig. 6i to show the quantification results with statistical analysis. The data clearly support our conclusion that CMTR2 knockout enhances sensitivity to indisulam treatment.

19. *Figure 5m/5n should have a Western blot for CMTR2. Maybe these are the same cells shown in Supplementary Figure 8a? If so, the authors need to say so. If not, they need to do additional Western blots.*

Response: Thank you for your valuable comment regarding the Western blot data. The cells used in Fig. 6n, o (Fig. 5m, n in the original submission) are the same as those shown in Supplementary Fig. 12a. We have referred to Supplementary Fig. 12a in the main text (page 9, line 318) to clarify this point.

20. *Fig. 6 is probably the weakest part of the paper at this point. The authors show a modest cooperative effect of the combination of indisulam and anti-PD1 antibodies and also some*

very modest degrees of responses to immune checkpoint blockade in CMTR2 mutant tumors. These data are intriguing and deserving of further development but probably not suitable for publication at this time.

Response: Thank you for your valuable feedback regarding Fig. 6 (Fig. 7 in the revised manuscript). We acknowledge that the synergistic effects of indisulam and anti-PD-1 combination therapy were modest and have therefore removed statements that overemphasized their cooperative effects from the manuscript. However, we believe that the core finding regarding CMTR2 deficiency and ICB sensitivity remains noteworthy. The LLC cell line is resistant to anti-PD-1 monotherapy, but CMTR2-knockout cells showed clear tumor growth suppression upon anti-PD-1 treatment, suggesting there is a meaningful biological relationship. Given that splicing abnormalities have recently gained attention as immunotherapy targets, we consider this result worthy of attention. To strengthen our findings, we have performed additional comparative analyses including CMTR2 WT patients as controls. These analyses revealed disease control rates of 90.9% (10/11) in CMTR2-mutant patients *versus* 68% (237/350) in WT patients, and durable clinical benefit rates of 64% (7/11) in CMTR2-mutant patients *versus* 46% (171/373) in WT patients. We have also identified an institutional case with both a truncating CMTR2 mutation and available ICB treatment data, who demonstrated a clinical response to pembrolizumab (anti-PD-1 antibody) treatment (Supplementary Fig. 21 and Supplementary Data 3). We acknowledge the limitations of this study, including the lack of experimental validation of neoantigen generation and the small sample size, which limits statistical power. We have explicitly stated these limitations in the Discussion section (page 12, lines 424–432). While we recognize these findings are exploratory, we believe they provide a valuable foundation for future research and potential therapeutic development in lung cancer patients and thus warrant inclusion with a description of appropriate caveats.

21. *In Discussion on page 9, lines 321-323. the authors state that involvement of CMTR2 in carcinogenesis was suggested in previous studies referencing several papers. The authors should include the reference to the study by Campbell et al, Nature Genetics 2016 that was the first one to identify CMTR2 / FTSJD1 as a recurrently mutated tumor suppressor gene in lung ADC in both introduction and discussion.*

Response: Thank you for your valuable suggestion. We have added the reference (Campbell et al. *Nature Genetics* 2016) in the Introduction (page 3, lines 84–85) and Discussion (page 11, lines 368–369) sections as you recommended. We acknowledge that this was the first study to identify *CMTR2/FTSJD1* as a recurrently mutated tumor suppressor gene in LADC.

22. *The authors should also discuss recent publications on CMTR2 including the recent evidence for adenosine modifications by CMTR2 as an alternative/addition to guanosine methylation (<https://pubmed.ncbi.nlm.nih.gov/39993262/>), the identification of CMTR2 polymorphisms as germline risk alleles for lung cancer (<https://pubmed.ncbi.nlm.nih.gov/39472694/>), and the evidence for a role of Cmtr2 in embryonic development*

(<https://pubmed.ncbi.nlm.nih.gov/39094818/>)

(<https://pubmed.ncbi.nlm.nih.gov/37436893/>)

Response: Thank you for your valuable suggestions regarding recent CMTR2 publications. We agree that these papers provide important insights into the function of CMTR2 and its role in cancer, which are crucial to understand CMTR2 mutations. We have incorporated all the suggested references as detailed below.

We have added the citation of Kück et al. *ChemBioChem* 2025 to our discussion of Cap2 RNA modification (page 5, lines 150–152) because it provides further support for the previously reported functions of CMTR2.

The identification of CMTR2 polymorphisms as germline risk alleles for lung cancer by Ivarsdottir et al. *Nature Genetics* 2023 offers compelling evidence that CMTR2 mutations are involved in carcinogenesis. We have cited this important study in the Discussion section (page 11, lines 378–379).

The studies by Dohnalkova et al. 2023 and Yermalovich et al. 2024 on the role of CMTR2 in embryonic development provide important evidence supporting its involvement in RNA metabolism and its critical biological functions. We have added these references to the Result section (page 5, lines 154–155).

23. *Throughout: The “NCI-H” cell lines are described as “H” cell lines. This should be corrected. For example, H3122 should be NCI-H3122, H1373 should be NCI-H1373 and so forth throughout the text.*

Response: Thank you for your helpful comment. We have corrected the cell line nomenclature throughout the entire manuscript and figures, changing “H” cell lines to “NCI-H” cell lines (“NCI-H3122” instead of “H3122”, “NCI-H1373” instead of “H1373”, and “NCI-H1915” instead of “H1915”) as you suggested.

Reviewer #2

“I co-reviewed this manuscript with one of the reviewers who provided the listed reports. This is part of the Nature Communications initiative to facilitate training in peer review and to provide appropriate recognition for Early Career Researchers who co-review manuscripts.”

Response:

We appreciate the additional insights provided through the co-review process. We have addressed the relevant comments in our responses to the individual reviewers.

Reviewer #3

Reviewer Comments:

1. *Mechanistic Link Between CMTR2 and Splicing Dysregulation*

The precise role of CMTR2 in splicing remains unclear. Given that CMTR2 primarily functions in mRNA capping, it is not immediately evident how its loss directly perturbs splicing. The authors should clarify whether the observed splicing defects arise from direct CMTR2 involvement or if secondary mechanisms (e.g., destabilisation of specific splicing regulators) are at play. Further functional assays could establish whether CMTR2 interacts with splicing machinery or indirectly affects splicing through changes in transcript stability.

Response: Thank you for this important comment regarding the mechanistic link between CMTR2 and splicing dysregulation. To address your specific concern, we have used three complementary approaches to investigate the relationship between CMTR2 deficiency and splicing alterations.

First, we investigated physical interactions between CMTR2 and splicing machinery components using immunoprecipitation experiments. We generated Flp-in T-REx 293 cells expressing FLAG-tagged WT CMTR2 or the K117N mutant (missense mutation of CMTR2 found in the characteristic splicing pattern cluster from the NCC cohort). Following FLAG immunoprecipitation, mass spectrometry revealed significant enrichment of U1 snRNP proteins in WT CMTR2 complexes, while the K117N mutant showed impaired U1 snRNP association (Fig. 5a–d, Supplementary Data 2). Western blotting confirmed the selective loss of U1 snRNP component binding to the K117N mutant (Fig. 5e). These results provide direct evidence for the physical interaction between CMTR2 and splicing machinery components, consistent with the findings from protein–protein interaction databases (Supplementary Tables 6 and 7).

Second, we examined the characteristics of splicing events affected by CMTR2 deficiency. The 71 shared SE events common to both human tumor samples and knockout cell lines exhibited GT-AG splice sites characteristic of major introns targeted by U1 snRNP (Supplementary Fig. 14). This finding supports the hypothesis that splicing alterations observed in CMTR2-deficient cells result from disrupted U1 snRNP function.

Third, we explored the molecular mechanism underlying this interaction using MD simulations. Our computational analysis revealed that missense mutations in CMTR2 significantly alter interactions with mRNA/snRNA (Fig. 2c, Supplementary Fig. 5). Specifically, mutations affecting catalytic residues involved in methylation increase the distance between CMTR2 and the second transcribed nucleotide of substrate mRNA/snRNA, indicative of weakened interactions, which provides a molecular basis for the disrupted enzymatic activity.

These experiments and simulation results connect with established literature regarding CMTR2 function. CMTR2 plays a crucial role in Cap2 methylation of snRNAs, with U1, U2, U4, and U5 snRNAs fully or nearly fully Cap2 methylated in various cell types (Krogh et al. *Org.*

Biomol. Chem. 2017). Importantly, CMTR2 knockout in HEK293T cells results in complete loss of Cap2 methylation of U1 snRNA (Despic et al. *Nature* 2023). Additionally, Cap2 modification of U2 snRNA is essential for spliceosome complex formation and RNA splicing (Dönmez et al. *RNA* 2004). Taken together, our findings suggest a mechanistic pathway: CMTR2 deficiency → loss of the interaction between CMTR2 and snRNP → loss of Cap2 methylation of snRNAs → destabilization of spliceosome complexes → RNA splicing anomalies.

Furthermore, the original manuscript contained evidence for secondary effects that likely contribute to the observed splicing dysregulation. Differential gene expression analysis of CMTR2-knockout cell lines revealed changes in expression of various splicing-related genes beyond snRNPs (Supplementary Fig. 15). Given that the splicing machinery forms a complex network, this suggests that altered interactions between CMTR2 and snRNPs have secondary effects on the broader splicing machinery.

Thank you for your valuable comment, which prompted us to investigate the relationship between CMTR2 deficiency and splicing alterations. Our results indicate that CMTR2 interacts with splicing machinery components and affects splicing through its role in Cap2 methylation. We have incorporated these findings into the Results (page 6, lines 195–202 and pages 8–9, lines 268–291) and Discussion (pages 11–12, lines 395–403) sections. To support these new findings, we have added detailed protocols to the Methods section in the sub-sections titled “Immunoprecipitation from Flp-in T-REx 293 cells” (page 19, lines 674–682), “NanoLC-MS/MS analysis” (pages 19–20, lines 684–691), and “Structural modeling and MD simulation of CMTR2 complexed with RNA and SAM” (pages 20–21, lines 710–736).

2. *Validation of Indisulam Sensitivity and Synthetic Lethality with RBM39*

The synthetic lethal relationship with indisulam is inferred based on RBM39 degradation. However, indisulam is known to have off-target effects. It would be valuable to demonstrate that RBM39 knockout (KO) alone recapitulates the CMTR2 loss phenotype, thereby confirming the specificity of this interaction.

Response: Thank you for this insightful comment. Upon examining the DepMap database of genetic dependencies in cancer cell lines, we found that *RBM39* is an essential gene for cell survival (Response only data Fig. 4). Complete knockout experiments would therefore be unable to show differences compared with control cells. Instead, we have evaluated this relationship by performing siRNA-mediated knockdown of *RBM39* in A549 control cells *versus* CMTR2-knockout cells.

RBM39 knockdown reduced proliferation of A549 CMTR2-knockout cells compared with control cells, demonstrating that CMTR2-deficient cells are more vulnerable to *RBM39* depletion (Supplementary Fig. 18). However, as correctly pointed out by the reviewer, sulfonamides exhibit pleiotropic effects and it is possible that cell proliferation is also inhibited through mechanisms that do not involve *RBM39*. We acknowledge that this is an

important consideration that could lead to identification of additional therapeutic targets. We have incorporated these revisions into the Results (page 9, lines 319–321) and Discussion (page 12, lines 411–414) sections.

Additionally, we have expanded the Methods section to include the detailed protocol for siRNA-mediated knockdown (pages 18–19, lines 645–653).

Response only data Fig. 4

A gene effect score of 0 defines a gene that is not essential, whereas a score of -1 corresponds to the median of all common essential genes.

3. *Data Analysis and Interpretation*

Short-Read Sequencing Limitations: The study relies on rMATS for AS event detection, which is constrained by the inherent limitations of short-read RNA sequencing. Long-read sequencing should be employed for validation, or at minimum, the authors should acknowledge the potential bias and incomplete detection of splicing events, especially those involving complex translocations.

Response: We appreciate your valuable comment regarding the limitations of short-read sequencing for alternative splicing analysis. While cost and technical constraints prevented us from reanalyzing all samples, we have performed long-read sequencing of A549 CMTR2-knockout cells to validate our findings. This additional analysis included visualization of read coverage and quantification of isoforms. We have added the results, which are consistent with our short-read sequencing data, to the Results section (pages 7–8, lines 253–256, Supplementary Fig. 10 a–g). Additionally, we have expanded the Methods section within the sub-section titled “RNA-seq” to include the detailed protocol for long-read direct RNA-seq (page 13, lines 454–460).

We are grateful for the Reviewer’s suggestion because this additional analysis has provided further support for the splicing changes induced by CMTR2 deficiency.

4. *tSNE Representation and Cluster Annotations:*

While the PSI-based tSNE captures histological features, the naming of clusters in Figure 1D may be misleading. The figure suggests a strict one-to-one relationship between clusters and features (e.g., splicing factor mutations, epithelial-mesenchymal transition [EMT] phenotype, neuroendocrine differentiation). However, feature expression within tSNE clusters is not comprehensively visualised.

Specific mutations (e.g., U2AF1, RBM10) appear in multiple locations, but only a subset is highlighted in Figure 1D. A more transparent depiction of feature distribution across the tSNE space would improve interpretability.

Response: Thank you for your insightful comments about our t-SNE representation and cluster annotations. To address your concerns, we have conducted a more detailed analysis of EMT status in clinical specimens by calculating EMT scores using the previously reported Kolmogorov-Smirnov test method based on EMT-related gene signatures. When visualized on the t-SNE landscape of the NCC cohort, samples with mesenchymal characteristics tended to be observed in a specific region and had a distribution distinct from samples with epithelial characteristics. However, as you correctly pointed out, it was inappropriate to highlight only certain samples. Therefore, we have removed Fig. 1d and instead included the complete distribution as Supplementary Fig. 2a–c. Consequently, we have also removed figures showing expression of CDH1 and VIM and the EMT enrichment analysis based on

selected EMT samples. We have modified the corresponding section in the main text (page 4, lines 126–130).

Similarly, regarding neuroendocrine differentiation, we have calculated enrichment scores using single-sample GSEA analysis with a previously published gene set. In addition to the figures showing the expression distribution of ASCL1 (a transcription factor involved in neuroendocrine differentiation) and SRRM4 (a splicing factor) that were already presented, we have added new Supplementary Fig. 2d, which shows the t-SNE landscape that reflects score information incorporating multiple genes related to neuroendocrine differentiation. We have updated the corresponding section in the main text (page 4, lines 131–135) to reflect these changes.

For U2AF1 and RBM10, we have also removed the figures highlighting only specific clusters and instead represented their complete distributions in Fig. 1d, e.

We have incorporated detailed descriptions of our EMT and neuroendocrine scoring methods into the Methods section (page 14, lines 493–502).

5. *Correction of Specific Claims:*

Line 97: "Other major histological subtypes displayed distinct distribution" should be revised, as minor subtypes are dispersed within the ADC cluster.

Line 115: "Gastric lineage" should be replaced with "gastrointestinal lineage."

Response: Thank you for your valuable suggestions. We have addressed your comments as detailed below.

Regarding Line 97, we have revised the text (page 3, lines 104–108) to more accurately reflect the distribution patterns observed in our plot as follows: “Adenocarcinoma (ADC, also referred to as LADC) and squamous cell carcinoma (SCC) tended to show different distributions. Some cases showed overlap between these two histological types, possibly representing combined-type samples with each differentiation component. Additionally, pleomorphic carcinoma was scattered throughout the landscape, reflecting its nature as a histological type that contains sarcomatoid and various histological subtype components”

As suggested, we have changed “gastric lineage” to “gastrointestinal lineage” (page 4, line 125).

6. *Methodology and Reproducibility*

The methodology is largely sound but should incorporate additional validation steps:

The accuracy of AS detection should be assessed using long-read sequencing, as this is now the standard for AS analysis.

The selection criteria for highly variable PSI events in PCA/tSNE should be explicitly stated to ensure unbiased dimensionality reduction.

Overall, the methods section provides sufficient detail for reproducibility.

Response: We greatly appreciate your thorough assessment of our methodology. Regarding your suggestion to incorporate long-read sequencing for AS detection validation, we have addressed this issue as mentioned in our response to Comment #3.

We appreciate your suggestion to clarify our selection criteria for highly variable PSI events. In the Methods section, we described the filters applied to select these events: average PSI values between 0.05 and 0.95, average sum read count ≥ 10 , percentage of samples with missing PSI values < 0.05 , and delta PSI values > 0.3 . For clarity, we have reorganized this description to explicitly state that these criteria were specifically designed to ensure unbiased dimensionality reduction for PCA/tSNE analysis by selecting events with sufficient read depth, minimal missing data, and high biological variability across samples. Thank you for acknowledging that our Methods section provides sufficient detail for reproducibility (pages 13–14, lines 468–476).

7. *Figure 1 Panel A (Workflow using rMATS)*

The reliance on short-read sequencing should be acknowledged as a limitation. While rMATS is unlikely to over-call splice variants, it may under-detect certain splicing events, particularly those involving trans-splicing or complex rearrangements.

Response: Thank you for your thoughtful comment regarding Fig. 1a. We agree that the reliance on short-read sequencing and rMATS for splicing analysis is a methodological limitation that should be acknowledged. As mentioned in our response to Comment #3, we have addressed this issue.

8. *Figure 1 Panel C (tSNE Clustering of LADC)*

Clarify how the PSI matrix was reduced to tSNE. Ensure that dimensionality reduction was performed in an unbiased manner (i.e., which PSI values were included in PCA?).

Response: Thank you for your valuable comment regarding the PSI matrix dimensionality reduction to t-SNE. Regarding panel C and your question about how the PSI matrix was reduced to t-SNE and ensuring unbiased dimensionality reduction, these details are described in the Methods section (pages 13–14, lines 468–476) as previously mentioned. For clarity, we have reorganized this description to explicitly state that our selection criteria were specifically designed to ensure unbiased dimensionality reduction for PCA/t-SNE analysis. The t-SNE plot shown in Fig. 1c was generated using SE type PSI values that met the following stringent selection criteria: average PSI values between 0.05 and 0.95, average sum read count ≥ 10 , percentage of samples with a missing PSI value < 0.05 , and delta PSI values > 0.3 . These criteria were implemented to select events with sufficient read depth, minimal

missing data, and high biological variability across samples, which effectively visualizes inter-tumor heterogeneity driven by distinct molecular subtypes.

9. *Panel D (PSI tSNE with Feature Enrichment)*

Demonstrate the actual distribution of splicing factor mutations, EMT markers, and neuroendocrine markers across the tSNE rather than relying solely on categorical cluster labels. Improve cluster labelling to reflect the heterogeneous nature of the data.

Response: Thank you for your valuable suggestion regarding panel d. As mentioned in our previous response, we have removed panel d and added new visualizations that display quantitative scores for EMT (Supplementary Fig. 2a–c) and neuroendocrine differentiation (Supplementary Fig. 2d–f) across the t-SNE landscape. We have updated the corresponding section in the main text (page 4, lines 126–130 and lines 131–135) to reflect these changes. This approach provides a more nuanced representation of the continuous distribution of these molecular features rather than relying on categorical cluster labels, which better reflects the heterogeneous nature of the data as you suggested.

10. *Supplementary Figure 1 Panel A (Clustering by PSI)*

Clustering appears robust, but the claim that "other major histological subtypes displayed distinct distribution" is questionable.

Response: Thank you for your insightful comment regarding Supplementary Fig. 1a. We agree with your assessment that our original claim that "other major histological subtypes displayed distinct distribution" needed reconsideration. As mentioned in our previous response, we have revised the text (page 3, lines 104–108) to more accurately reflect the distribution patterns of each histological subtype. This modification provides a more nuanced and precise description that aligns with the actual clustering patterns observed in our analysis.

11. *Supplementary Figure 1 Panel G (GSEA of GI-related pathways)*

Revise "gastric lineage" to "gastrointestinal lineage."

Response: Thank you for your comment regarding panel g. As addressed in our response to Comment #5, we have changed "gastric lineage" to "gastrointestinal lineage" (page 4, line 125). This modification provides a more accurate description of the lineage identity observed in our analysis.

12. *Supplementary Figure 2 Panels A-D (EMT Phenotype Validation)*

Boxplots alone do not fully demonstrate the spatial distribution of EMT features. Overlaying EMT markers on the tSNE plot would provide stronger support for this claim.

Panel D (GSEA for EMT Cluster)

Further validation of EMT assignment within tSNE is warranted.

Response: Thank you for your valuable comments regarding Supplementary Fig. 2a–d and the spatial distribution of EMT features. As previously mentioned, we have revised our approach to visualize the distribution of EMT cases across the splicing landscape. We have removed the boxplots and presented a visualization of the spatial distribution of a comprehensive EMT score (Supplementary Fig. 2a–c) that incorporates expression information for multiple EMT-related genes. We have updated the corresponding section in the main text (page 4, lines 126 to 130) to reflect these changes. This approach provides a more intuitive visualization of how EMT features are distributed across the t-SNE landscape and addresses your concerns about the need for further validation of EMT assignment within the t-SNE plot.

13. *Supplementary Figure 2 Panels E-F (Neuroendocrine Gene Expression)*

Similar to EMT analysis, consider mapping neuroendocrine markers onto the tSNE plot.

Response: Thank you for your suggestion. As previously mentioned in our response, we have modified our approach to visualize the distribution of neuroendocrine differentiation across the splicing landscape. Similar to our EMT analysis, we have presented a visualization of the spatial distribution of a comprehensive neuroendocrine score (Supplementary Fig. 2d) that incorporates expression information for multiple neuroendocrine-related genes. We have updated the corresponding section in the main text (page 4, lines 131–135) to reflect these changes. This approach provides a more intuitive visualization of how neuroendocrine features are distributed across the t-SNE landscape, addressing your recommendation to map neuroendocrine markers onto the t-SNE plot.

Reviewer #4

Reviewer Comments:

1. *The authors identify CMTR2 as a putative tumour suppressor but there is no direct analysis of the effects of CMTR2 deletion on tumour cell growth. Can the authors investigate whether CMTR2 deletion increases tumorigenic phenotypes in vitro and in vivo?*

Response: Thank you for this important suggestion. In response to this comment, we have conducted additional functional experiments to investigate the effects of CMTR2 deletion on tumorigenic phenotypes both *in vitro* and *in vivo*.

For *in vitro* analysis, we have performed colony formation assays comparing A549 CMTR2-knockout cells with control cells (n=6 technical replicates) (Response only data 5a). For *in vivo* experiments, we have transplanted A549 CMTR2-knockout and control cells into immunodeficient mice (Response only data 5b) and transplanted LLC CMTR2-knockout and

control cells into syngeneic C57BL/6 mice (Response only data 5c) to measure tumor growth (n=8 mice per group).

However, neither experimental system demonstrated increased tumor growth with CMTR2-knockout cells, and we were unable to observe tumorigenic phenotypes associated with CMTR2 deficiency in our model systems.

Despite these results, previous studies have suggested that CMTR2 is involved in carcinogenesis. For instance, tumorigenesis studies using genetically engineered mouse models that induce CMTR2 inactivation showed that loss of CMTR2 promotes proliferation of KRAS-driven lung cancer cells (Cai et al. *Cancer Discovery* 2011, ref. 55). Additionally, germline loss-of-function variants of CMTR2 were reported to be significantly associated with an increased risk of LADC and cutaneous melanoma (Ivarsdottir et al. *Nature Genetics* 2024, ref. 56), suggesting that *CMTR2* functions as a tumor suppressor gene.

We propose that the discrepancy between our results and those of previous reports may be attributable to the experimental model used. Our knockout experiments represent a model in which CMTR2 is inactivated after cancer development rather than during the cancer initiation process. This temporal difference in CMTR2 inactivation may explain why we did not observe the expected tumorigenic phenotypes.

Therefore, we conclude that while our model of CMTR2 knockout in established cancer cell lines was highly suitable for observing the resulting splicing abnormalities (a primary focus of our study), this experimental system is not necessarily appropriate for demonstrating the tumor-suppressive functions in relation to tumor proliferation. We have cited the aforementioned studies that evaluated CMTR2 in the carcinogenesis process in the Discussion section (page 11, lines 378–379).

Response only data Fig. 5

a

b

c

2. This is particularly relevant given the immunogenic phenotype following CMTR2 deletion, which would be expected to suppress tumour initiation and growth. What is the proposed mechanism of action of CMTR2 deletion?

Response: Thank you for your important question regarding the relationship between CMTR2 deletion and the immunogenic phenotype. In our study, we demonstrated that CMTR2 deletion causes RNA splicing abnormalities, leading to transcriptional alterations. These splicing abnormalities potentially contribute to increased neoantigen production. This phenomenon resembles cancers with a high tumor mutation burden (TMB). Generally, high-TMB cancers theoretically produce numerous neoantigens and should be highly

immunogenic, but still manage to evade immune elimination and proliferate. This phenomenon of evading immune elimination despite a high TMB and high immunogenicity is one of the important paradoxes in cancer immunology. Similar mechanisms may operate in CMTR2-deficient tumors. Specifically, we consider the following possibilities:

1. Acquisition of immune evasion mechanisms

CMTR2-deficient tumors may acquire certain immune evasion mechanisms that allow them to avoid immune elimination despite increased production of neoantigens. Additionally, chronic immune responses sometimes cause immune exhaustion, which may also be involved.

2. Involvement in cell proliferation

Cap2 methylation abnormalities due to CMTR2 deficiency may cause changes in intracellular signaling that are advantageous for cell proliferation or survival, either through splicing changes or other mechanisms. Previous research (Cai et al. *Cancer Discovery* 2011, ref. 55) showed that CMTR2 deletion promotes tumor growth, although the detailed molecular mechanisms have not been fully elucidated. These changes might provide proliferative advantages that outweigh the disadvantages of increased immunogenicity.

Interestingly, our study showed that CMTR2-deficient tumors respond favorably to ICB therapy. This suggests that CMTR2-deficient tumors rely on immune evasion mechanisms and that inhibition of these mechanisms can elicit therapeutic effects. This is consistent with the observation that a high TMB is often a good predictor of response to ICB therapy. Therefore, our central hypothesis is that while loss of CMTR2 creates an immunogenic vulnerability, it simultaneously provides pro-survival advantages and drives upregulation of compensatory immune evasion pathways, allowing the tumor to ultimately grow and persist. However, further research is needed to elucidate the specific immune evasion mechanisms and their molecular basis in CMTR2-deficient tumors.

We have addressed these mechanistic considerations and research limitations in the Discussion section (page 12, lines 424–429), where we acknowledge that detailed investigation of Cap2 alterations and their functional consequences are crucial to understand how CMTR2 influences tumor progression and immune responses.

3. *In Figure 4, the authors investigate splicing changes induced by loss of CMTR2. Are any functional processes enriched in these altered splicing events? And could these indicate potential mechanism of CMTR2 deletion driving tumourigenesis?*

Response: Thank you for your insightful question regarding the functional processes potentially enriched among the altered splicing events caused by CMTR2 deletion. We have performed enrichment analysis of the overlapping differentially spliced events/genes (71 events/59 genes) identified in CMTR2-mutated cases from the NCC and TCGA cohorts as well as of our CMTR2-knockout model cells. We have used multiple gene set collections including

GO Biological Process, GO Molecular Function, and KEGG pathways. However, our analysis did not reveal significant enrichment of gene sets that would suggest a specific mechanism by which CMTR2 deletion promotes tumorigenesis (Response only data Fig. 6).

While the splicing changes induced by CMTR2 deletion are not random and occur at specific splicing junctions, our findings suggest they are not function-specific but rather sequence-specific. This suggests that CMTR2 influences splicing based on specific genomic sequences rather than targeting genes involved in particular functional pathways. This sequence-specific nature of CMTR2-mediated splicing regulation may explain why conventional functional enrichment analysis of the affected splicing events did not yield clear insights into the mechanisms underlying tumorigenesis.

Response only data Fig. 6

4. *Can the phenotypic effects in the PDO model (Figure 5), be rescued by reintroducing wild-type CMTR2?*

Response: Thank you for asking this important question. We attempted to perform rescue experiments by reintroducing WT CMTR2 into CMTR2-deficient PDOs. However, we were unable to establish stable CMTR2-expressing organoids due to technical limitations. While we acknowledge that PDO rescue experiments would provide additional validation, we believe our current data provide robust evidence for the causal relationship between CMTR2 status and indisulam sensitivity. Specifically, we successfully demonstrated rescue effects in cell line models (Fig. 6n, o and Supplementary Fig. 17f) and observed consistent indisulam/E7820 sensitivity in multiple CMTR2-deficient models including both cell lines and organoids. We plan to address the technical limitations of PDO genetic manipulation in future studies.

5. *In Fig 6F the authors conclude that CMTR2 mutation is predictive of a good response to immune checkpoint blockade but only the CMTR2 mutated samples are shown. What is the responsiveness in CMTR2 non-mutated tumours so an accurate comparison of response rate can be carried out?*

Response: Thank you for your important question regarding the comparison of responsiveness between CMTR2-mutated and non-mutated tumors. We have analyzed the efficacy of immune checkpoint inhibitors in patients without CMTR2 truncating mutations as suggested. Although we could not demonstrate statistical significance due to the small number of samples with CMTR2 truncating mutations, we consistently observed favorable trends in the CMTR2 truncating mutation group in terms of the disease control rate, proportion of durable clinical benefit, and PFS. We have revised the corresponding section of the manuscript (pages 10–11, lines 355–364) to include this comparison.

Reviewer #6

“I co-reviewed this manuscript with one of the reviewers who provided the listed reports. This is part of the Nature Communications initiative to facilitate training in peer review and to provide appropriate recognition for Early Career Researchers who co-review manuscripts.”

Response:

We thank both reviewers for their combined efforts and hope we have addressed all their concerns in the responses provided above.

[Point-by-point responses to the reviewers]

“Mutation of CMTR2 in Lung Adenocarcinoma Alters RNA Alternative Splicing and Reveals Therapeutic Vulnerabilities”

Authors: Nukaga et al.

We thank the editor and reviewers for their careful review of our revised manuscript, and for their valuable feedback. We are grateful for the opportunity to further improve our work through this revision process. Below, we provide point-by-point responses to the comments made by the Reviewer. All changes to the manuscript are highlighted in yellow.

Reviewer #3

Reviewer Comments:

We would like to acknowledge the considerable work that has gone into addressing the earlier concerns. Overall, the revisions are thorough and have significantly strengthened the manuscript. The concern about the lack of mechanistic insight into how truncating mutations in CMTR2 lead to splicing alterations has been addressed particularly well. The new data showing that CMTR2 physically interacts with components of the splicing machinery, and that this interaction is lost when this truncated, provide a much clearer picture of its role. The accompanying mRNA-level confirmation of splicing effects further supports the proposed mechanism and adds weight to the biological relevance of the findings.

Regarding validation of the splicing changes using long-read sequencing: although the ONT direct RNA sequencing results have not been incorporated into a main figure, it is good to see that the authors have performed this work. The relevant details (lines 253–256 and SI Fig. 10) suggest the long-read data generally support the conclusions drawn from short-read analyses. The BAMBU analysis of target genes indicating differential isoform expression following CMTR2 knockout is in line with expectations. However, a direct comparison between short- and long-read data—ideally side-by-side—would have been helpful in fully evaluating concordance between these approaches. The shortcomings of not using long-read sequencing throughout the study should be discussed in terms of false positives and limitations to this work within the discussion.

Response:

We sincerely appreciate the reviewer's thorough evaluation of the revised manuscript, as well as the constructive feedback. We are pleased that the reviewer acknowledges the substantial improvements to the revised version, particularly regarding the mechanistic insights into the role of CMTR2 in splicing regulation.

Regarding presentation of long-read sequencing data, we appreciate the reviewer's suggestion that we include a direct side-by-side comparison with the short-read data. We

would like to clarify that our decision to present the long-read sequencing results in Supplementary Figure 10 rather than in a main figure was made to maintain clear and coherent data presentation. Given that the long-read sequencing analyses encompass multiple aspects (long-read direct RNA-seq coverage plots, isoform detection by BAMBU analysis, and isoform quantification), we believed that consolidating these related analyses into a dedicated supplementary figure would provide a more comprehensive and organized view of the long-read data. Additionally, space constraints with respect to the content of the main figures necessitated this arrangement. Nevertheless, we acknowledge that this presentation format may have made the direct comparison between short- and long-read approaches less immediately apparent.

To address the reviewer's important point about discussing the limitations of our sequencing approach, we have revised the Discussion section (page 12, lines 403–410) as follows:

"Our splicing analysis, conducted primarily through short-read RNA-seq, revealed characteristic splicing abnormalities associated with CMTR2 deficiency; however, short-read-based tools have inherent limitations, including potential false positives and false negatives, with respect to resolving complex splicing events such as trans-splicing, as well as quantifying accurate isoforms. Although we validated key splicing events in CMTR2-knockout cells using long-read sequencing, which showed strong concordance with the short-read data obtained from SPPL2A and SMARCA1, comprehensive genome-wide analysis of the long-read sequencing data across multiple samples was not feasible. Future systematic long-read sequencing analyses would allow more comprehensive characterization of splicing dysregulation and complexity driven by CMTR2-deficiency."

We believe these additions acknowledge the limitations while at the same time maintaining the validity of our core findings, which are well-supported by both sequencing approaches.